# Comparative multi-omic analysis reveals conserved and derived mechanisms of fin and limb regeneration

Josane F. Sousa[1,2], Gabriela Lima[1,2], Louise Perez [1], Hannah Schof [1] & Igor Schneider [1] ✉

Comparative studies of vertebrate appendages offer a powerful framework for uncovering shared components of an ancestral regeneration toolkit. Here, we employed a multi-omics comparative approach leveraging the regenerative capacity of the axolotl, zebrafish, and *Polypterus senegalus*, a fish capable of full fin regeneration. We identified conserved markers of proximal and distal blastema territories, shared activation of DNA damage repair, *hif1a*-mediated hypoxia response, and sequential activation of pro- and anti-inflammatory program. Apical epithelial ridge markers were expressed in both the wound epidermis and distal mesenchyme during limb and fin regeneration. Notably, *hif4a*-expressing erythrocytes were uniquely associated with proximal limb and fin amputations but not fin rays, while epidermal myoglobin expression was upregulated only in *Polypterus* and zebrafish fins. Genome-wide chromatin profiling identified candidate regeneration-responsive elements and a conserved enrichment for AP-1 transcription factor binding. Together, these findings identify shared and derived mechanisms of limb and fin regeneration.

Among limbed vertebrates (tetrapods), salamanders are the only clade capable of regrowing severed limbs as adults. This regenerative capacity may have been inherited by tetrapods from their fish ancestors, retained in modern salamanders, partially preserved in frogs (limited to larval stages), but entirely lost in amniotes (Fig. 1a). Supporting this hypothesis, fossil evidence suggests that limb regeneration predates the origin of stem salamanders[1,2]. A phylogenetic survey of complete paired fin regeneration—defined as regeneration following amputation at the endoskeleton-bearing proximal fin region—demonstrated that this ability is present across species representing all major bony fish clades[3,4], including zebrafish larvae[5,6]. Furthermore, decades of comparative studies using the zebrafish caudal fin as a model system have revealed commonalities between limb and fin ray regeneration programs[7], including cell-type-specific genetic and epigenetic programs[8]. Altogether, these findings support a shared evolutionary origin of limb and fin regeneration, underscoring the importance of comparative studies as a powerful framework for identifying the core cellular and molecular components underlying vertebrate appendage regeneration.

While the adult zebrafish serves as a valuable model for comparative and functional studies in fin regeneration, its regenerative capacity is limited to the distal fin rays[5] (Fig. 1c), a structure that lacks direct homology with limbs[9] and the cellular complexity and diversity observed during limb regeneration[8]. Additionally, the teleost-specific whole-genome duplication (WGD) event presents challenges for genetic comparisons between zebrafish and salamanders, complicating the identification of conserved regenerative mechanisms[10,11]. To address these limitations, we have established the Senegal bichir, *Polypterus senegalus*, as a model, leveraging its remarkable regenerative abilities. *Polypterus*, a non-teleost ray-finned fish (actinopterygian), diverged from teleosts prior to the teleost-specific WGD (Fig. 1a), maintaining a 1:1 gene orthology with tetrapods[12]. Unlike zebrafish, *Polypterus* can readily regenerate entire fins as adults, including the proximal fin region, which contains endoskeletal

[1]Department of Biological Sciences, Louisiana State University, Baton Rouge, LA, USA. [2]These authors contributed equally: Josane F. Sousa, Gabriela Lima. ✉e-mail: igors@lsu.edu

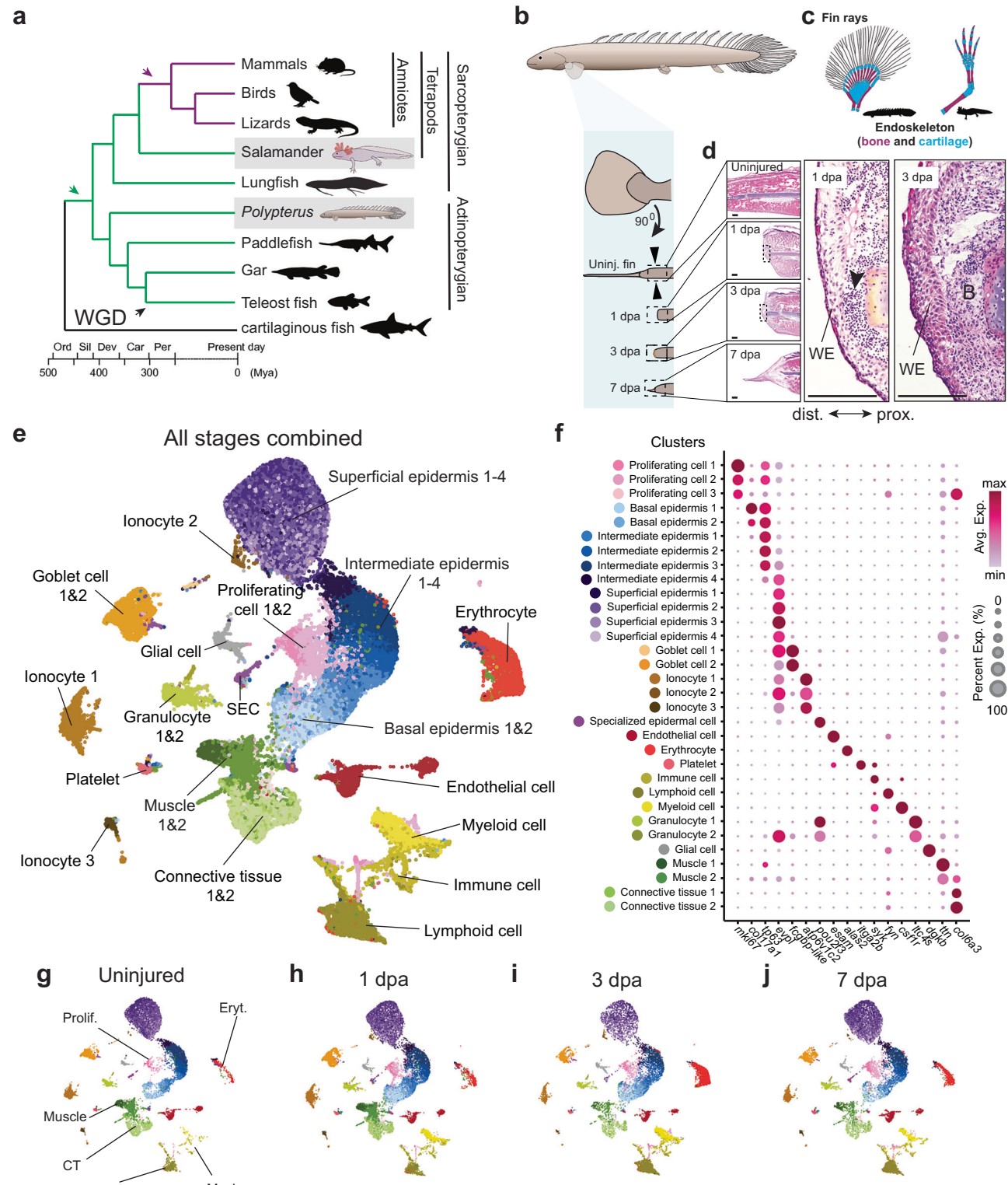

**Fig. 1 | SnRNA-seq uncovers cellular diversity of *Polypterus* fin at homeostasis and regrowth. a** Vertebrate phylogeny showing absence (black line), presence (green line), or loss (magenta line) of paired appendage regeneration; green arrow indicates gain, magenta arrow indicates loss of regenerative capacity, and black arrow indicates whole genome duplication (WGD) event. **b** *Polypterus* pectoral fin; black arrowheads denote amputation site. **c** *Polypterus* fin and axolotl limb skeleton; rays (black) and endoskeletal elements (blue and red). **d** Histological sections of stages assayed in this study; black arrowhead in 1 dpa indicates immune cell infiltration. **e** UMAP plot of nuclei from all stages combined (uninjured, 1 dpa, 3 dpa, and 7 dpa) identified 32 distinct clusters. **f** Dot plot of representative gene markers of major cell types. **g**–**j** UMAP plots showing changes in cell populations across uninjured and regenerating stages. Scale bars = 100 µm. WE wound epidermis, B blastema. Proximal (prox.) distal (dist.) axis indicated in (**d**).

elements sharing deep homology with limb bones[13,14] (Fig. 1c). Additionally, the proximal fin region contains organized muscle masses, diverse connective tissue populations such as tendon and ligament fibroblasts, and fibrocartilaginous joints, contributing to its overall cellular complexity. Proximal fin amputations thus enable the assessment of regenerative processes in a cellularly intricate context, including interactions among muscle, tendon, connective tissue, cartilage precursors, vasculature, and nerves. Furthermore, our previous studies identified signaling pathways commonly upregulated in both salamanders and *Polypterus*[4], highlighting its potential as a model for evolutionarily informed comparative studies of vertebrate appendage regeneration.

To comprehensively identify the core cellular, genetic, and gene regulatory components of *Polypterus* fin regeneration, we employed single-nucleus RNA-sequencing (snRNA-seq) and spatial RNA-sequencing (spatial RNA-seq). Additionally, we generated spatial RNA-seq datasets for axolotl limb regeneration and leveraged publicly available single-cell RNA-sequencing (scRNA-seq) data from axolotl limb[15] and snRNA-seq data from zebrafish caudal fin[8] regeneration. This approach enabled a broad comparative analysis of limb and fin regeneration during homeostasis and across successive regeneration stages. Our findings revealed conserved and derived features of limb and fin regeneration, including immune cell recruitment, infiltration of *hif4a*+ erythrocytes, blastema cell heterogeneity, wound epidermis and blastema gene expression profiles, proximal-distal subdivision of connective tissue cells, sequential activation of pro- and anti-inflammatory genetic programs, DNA damage response, and hypoxia-adaptive processes. Finally, using the assay for transposase-accessible chromatin with high-throughput sequencing (ATAC-seq), we uncovered candidate regeneration-responsive *cis*-regulatory elements and identified AP-1 transcription factors as potential key conserved regulators in the early response to fin and limb amputation.

## Results

### The cellular landscape of *Polypterus* fin regeneration
To assess fin regeneration at the single-cell level, we selected timepoints representative of homeostasis (the uninjured proximal segment of the pectoral fin) and regeneration at 1, 3, and 7 days postamputation (dpa) (Fig. 1b). Fin amputation at the level of the endoskeletal elements resulted in complete wound closure and presumptive immune cell infiltration at 1 dpa (Fig. 1d). At 3 dpa, the wound epidermis was multilayered, containing distinct basal and suprabasal layers, and subjacent mesenchymal cells forming the presumptive blastema. At 7 dpa, the wound epidermis extended distally, forming a dorso-ventrally constricted outgrowth in which fin rays will form.

Using combinatorial barcoding technology (Parse Biosciences), we profiled almost 63,000 nuclei from fins at homeostasis and during regeneration in biological replicates. Trailmaker (Parse Biosciences, 2024) was used to process and analyze snRNA-seq data (Supplementary Fig. 1a). Unbiased clustering of all nuclei from all timepoints revealed 32 transcriptionally distinct populations (clusters) (Fig. 1e and Supplementary Fig. 1b). Cluster annotation based on transcriptional profiles identified multiple cell populations, including proliferating cells, muscle, connective tissue, glial cells, platelets, erythrocytes, endothelial cells, various immune cells and epidermal cells of multiple types (basal/intermediate/superficial epidermis, goblet cells and ionocytes) (Fig. 1e, f, Supplementary Figs. 1b and 2). We detected three populations of proliferating cells (Fig. 1e, f and Supplementary Fig. 2a, b). Proliferating cell clusters 1 and 2 presented epithelial cell features (expression of *tp63* and *evpl*), whereas expression of *col6a3* in proliferating cell 3 suggests that this cluster constitutes connective tissue proliferating cells.

To assess changes in cell populations and gene expression patterns, we evaluated cell clusters across regeneration stages. At 1 dpa, the major changes observed were the increases in proliferating cells,

myeloid cells, lymphoid cells, and basal and intermediate epidermal cells (Fig. 1g, h and Supplementary Fig. 2c), consistent with wound closure completion and immune cell infiltration observed via histology (Fig. 1d). At 3 dpa, all immune cell types were increased relative to 1 dpa, an indicative of substantial immune cell infiltration, yet the most notable change was the large increase in erythrocytes (Fig. 1g–j and Supplementary Fig. 2c). Massive immune cell infiltration, especially macrophages, and the formation of erythrocyte clumps in the limb blastema, have both been identified as features of salamander limb regeneration[16,17]. Finally, we observed reduction of muscle, connective tissue, and terminally differentiated epidermal cell (ionocytes, goblet cells, and superficial epidermis) clusters at 3 dpa, and a reversal of this trend at 7 dpa (Fig. 1g–j and Supplementary Fig. 2c).

### Shared spatial gene expression domains of fin and limb regeneration
To complement our snRNA-seq dataset and enable comparisons of gene expression territories during *Polypterus* fin and axolotl limb regeneration, we generated high throughput, spatially resolved gene expression data using Visium spatial transcriptomics technology (10x Genomics). In both species, a regeneration series was assayed, with the goal of capturing homeostasis, wound closure, wound epidermis formation, and blastema establishment. To this end, *Polypterus* fins were sampled at homeostasis (proximal segment of uninjured pectoral fin), 1 dpa, 3 dpa, and 7 dpa (Fig. 2a)—matching the stages assayed via snRNA-seq. Axolotl limb regeneration was assessed at homeostasis (uninjured forearm segment), 3 dpa, 7 dpa, and 14 dpa (Fig. 2e)—matching stages reanalyzed from publicly available scRNA-seq data from axolotl limb[15] (Supplementary Fig. 3a).

We applied k-means clustering to identify major tissue compartments across spatial transcriptomic samples. To determine the optimal number of clusters for k-means clustering, we generated elbow plots of within-cluster sum of squares (WCSS)[18] across $k = 2$–10 (Supplementary Fig. 4). A sharp reduction in WCSS was observed from $k = 2$ to 4, with diminishing improvement, thereafter, indicating $k = 4$ as a suitable balance between cluster compactness and model complexity. Clustering identified tissue types and orthologous markers for epidermis (*agr2*, suprabasal layers), muscle (*ckm*), skeleton (*col2a1*), connective tissue (*col6a3*) and immune cells (*mmp3*) in the *Polypterus* fin (Fig. 2b) and axolotl limb (Fig. 2f). Furthermore, we assessed the expression of these markers in their corresponding cell types in our *Polypterus* snRNA-seq dataset and in the scRNA-seq data from axolotl limbs. Our analysis confirmed matching expression of these markers in clusters corresponding to epidermis, skeleton (a small subset of the connective tissue cluster), muscle, connective tissue and a subpopulation of myeloid cells (Fig. 2c, d, g, h), validating the combined analysis of our spatially and cellularly resolved gene expression datasets in the downstream comparisons described in subsequent sections. The combination of *Polypterus* snRNA-seq and spatial RNA-seq data allowed us to better refine cell cluster annotation and identify subsets of cell types within annotated clusters. This was especially relevant for the identification of the fin skeletal compartments. As described above, skeletal cells were found to correspond to a subset of connective tissue cells, as evidenced by the expression of the marker *col2a1*, detected via spatial RNA-seq in the *Polypterus* fin skeleton (Fig. 2b–d).

### Distinct regeneration programs of endochondral and dermal skeleton in *Polypterus*
Complete fin regeneration in *Polypterus* requires regrowth of both the endochondral skeleton and the dermal fin rays, providing an opportunity to distinguish the cells and genetic programs that contribute to each process. To this end, we examined canonical markers of dermal and endochondral regeneration in our snRNA-seq and spatial RNA-seq datasets.

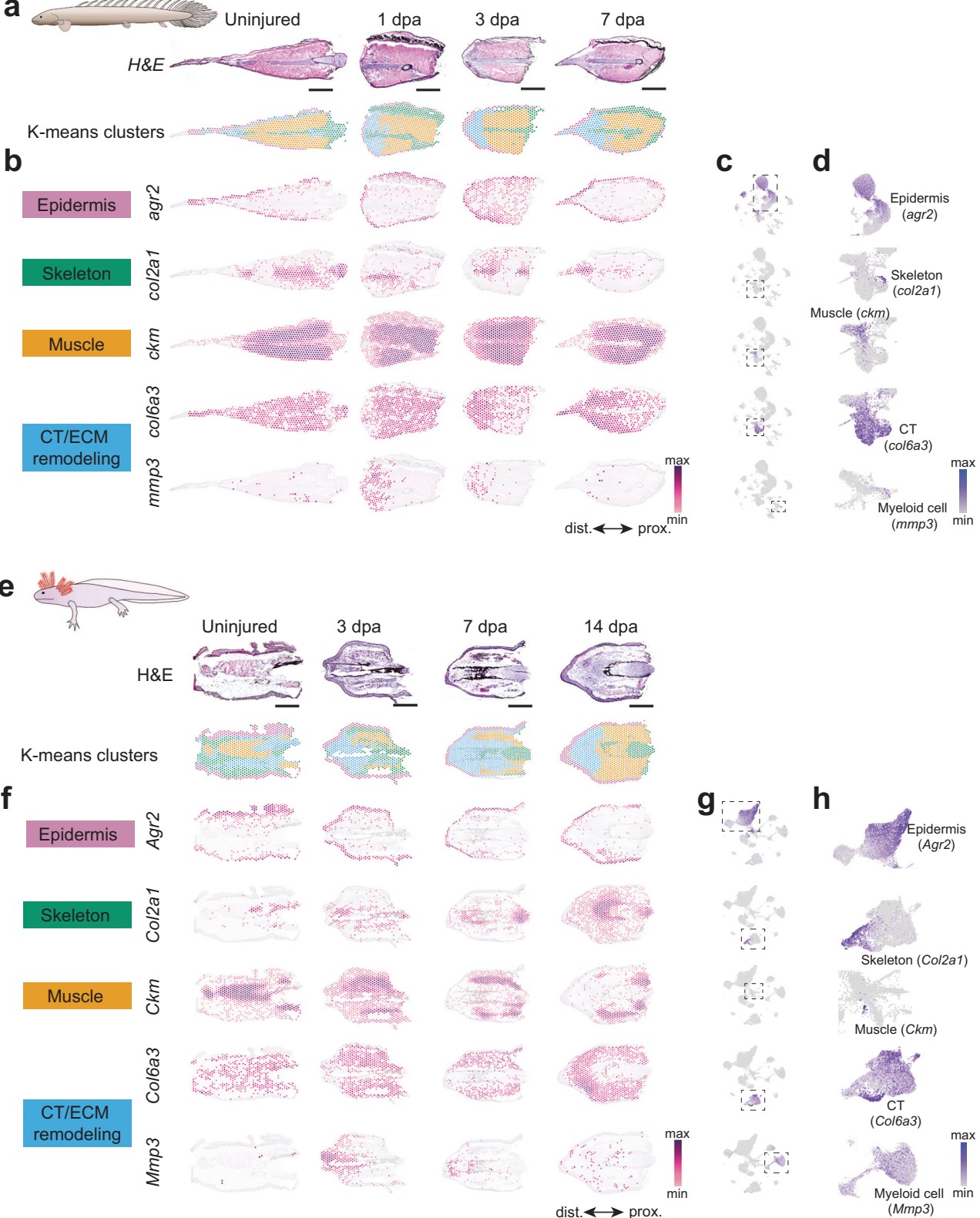

**Fig. 2 | Spatial transcriptomics identifies conserved tissue compartments in *Polypterus* and axolotl across homeostasis and regeneration stages.** Histological sections used for spatial transcriptomics of *Polypterus* fin (**a**) and axolotl limb (**e**). **b**, **f** Unbiased k-means clustering identifies gene expression territories corresponding to the epidermis, skeleton, muscle, and connective tissue (CT)/Extracellular Matrix (ECM) remodeling. Spatial expression profile of gene markers of epidermis (*agr2*), skeleton (*col2a1*), muscle (*ckm*), and CT/ECM (*col6a3*, *mmp3*, respectively) in *Polypterus* (**b**) and axolotl (**f**). UMAP plots of all stages combined showing expression of *agr2*, *col2a1*, *ckm*, *col6a3*, and *mmp3* in cell clusters in *Polypterus* (**c**, **d**) and their orthologs in axolotl (**g**, **h**); regions denoted by boxes with dashed lines in **c** and **d** are expanded in **g** and **h**, respectively. Scale bars in **a** and **e** = 1 mm. Proximal (prox.) distal (dist.) axis indicated in **b** and **f**.

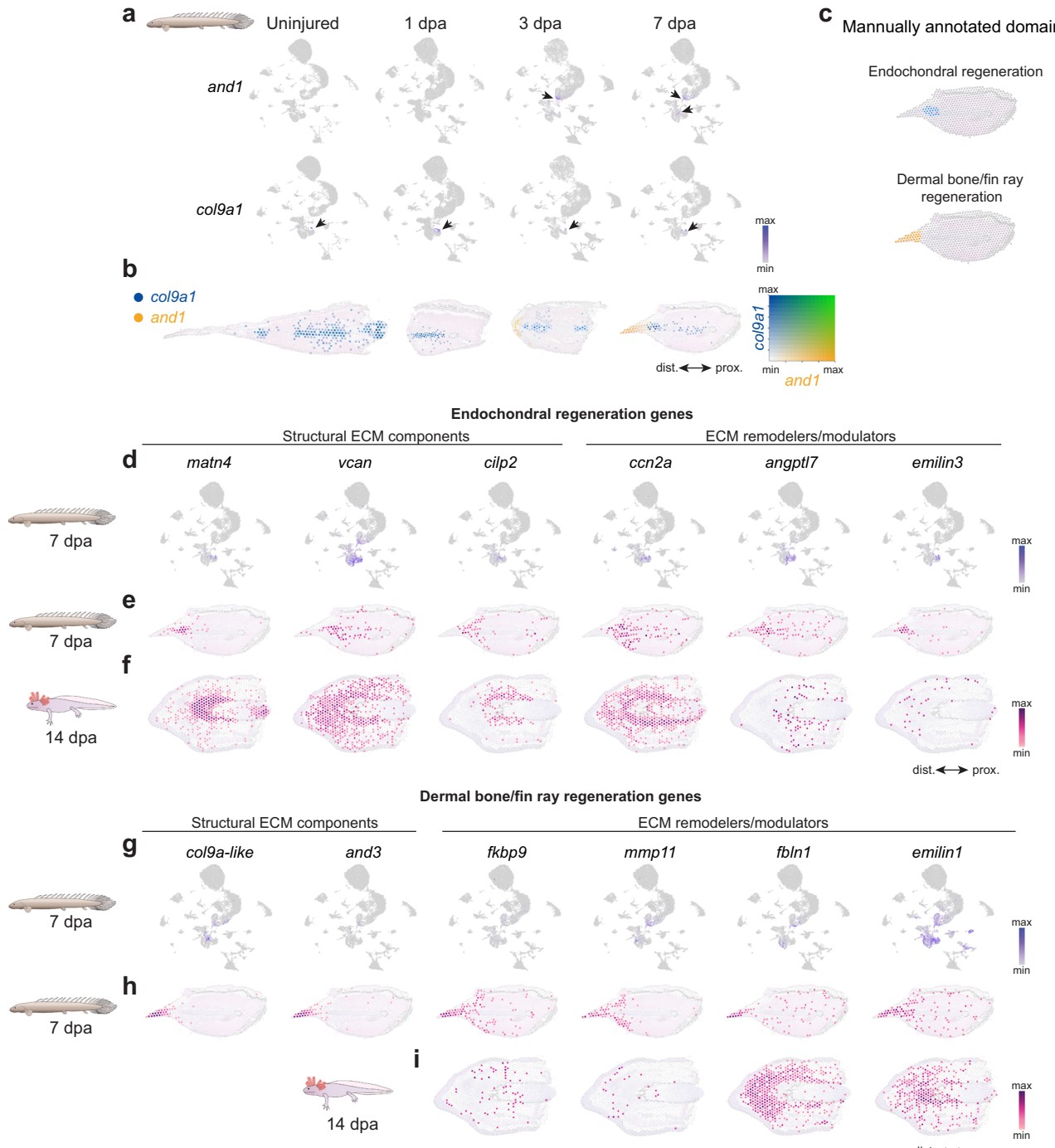

**Fig. 3 | Spatial transcriptomics reveals transcriptionally distinct domains of endochondral and fin ray regeneration. a** UMAP plot showing *and1* and *col9a1* expression across uninjured and regenerating stages; arrowheads indicate marker expression within clusters. **b** Spatial transcriptomics showing *col9a1* (blue) and *and1* (yellow) expression domains across uninjured and regenerating stages. **c** Manually annotated domains in spatial transcriptomics showing endochondral regeneration (top) and dermal bone regeneration (bottom) territories at 7 dpa. UMAP plots of *Polypterus* at 7 dpa showing expression of endochondral regeneration-associated genes (**d**), and spatial expression at 7 dpa in *Polypterus* (**e**) and 14 dpa in axolotl (**f**). UMAP plots of *Polypterus* at 7 dpa showing expression of dermal bone regeneration-associated genes (**g**), and spatial expression at 7 dpa in *Polypterus* (**h**) and 14 dpa in axolotl (**i**). ECM, Extracellular Matrix. Proximal (prox.) distal (dist.) axis indicated in (**b**, **f**, **i**).

One of the best characterized markers of dermal skeleton development and regeneration is *and1*[19,20]. Consistent with observations in zebrafish fin development[19], we detect the onset of *and1* expression in nuclei of basal epidermal cells of the *Polypterus* fin at 3 dpa, in addition to its canonical expression in mesenchymal progenitors at 7 dpa (Fig. 3a, upper panel). Conversely, *col9a1* – a marker of chondrocytes of endochondral growth plates in fish and mammals[21,22] – was expressed in a discrete subpopulation of connective tissue cells at homeostasis and throughout regeneration stages (Fig. 3a, lower panel). Spatial analysis confirmed that *and1* expression marks the distal-most fin ray regeneration territory, while *col9a1* was expressed throughout the cartilaginous endoskeleton and immediately subjacent to the *and1* domain at 3 dpa and 7 dpa (Fig. 3b).

To identify gene expression signatures associated with these territories, we manually defined regions of interest at 7 dpa corresponding to fin ray regeneration (*and1*+) and endochondral regeneration (*col9a1*+) (Fig. 3c). Analysis of the top 50 genes differentially expressed genes revealed distinct extracellular matrix (ECM) components enriched in each compartment (Supplementary Data 1). In our snRNA-seq dataset, structural ECM genes such as *matn4* and *cilp2* were specifically expressed in the connective tissue cell subpopulation corresponding to *col9a1*+ endochondral cells, whereas expression of *vcan* was also detected in other connective tissue cells and basal epidermis (Fig. 3d). In addition, the ECM remodelers *ccn2a*, *angptl7*, and *emilin3* were expressed in connective tissue cells that overlapped with the *col9a1*+ endochondral population. Spatial RNA-seq analysis confirmed the expression of these ECM components and remodelers within and adjacent to the endochondral regeneration domain (Fig. 3e). By analyzing our spatial transcriptomics dataset of the regenerating axolotl limb at 14 dpa, we showed that these genes also marked the presumptive endoskeleton regeneration front in the axolotl, immediately distal to and surrounding the endoskeleton in the stump, except for *emilin3*, which was expressed at low levels in scattered distal spots (Fig. 3f).

In contrast, the fin ray regeneration domain at 7 dpa was characterized by expression of dermal ECM genes, including *and3* and *col9a1-like*, a paralog of *col9a1* absent from tetrapod genomes, which was expressed in subsets of basal epidermis and connective tissue cells (Fig. 3g). Spatial RNA-seq confirmed that the expression of these markers was largely restricted to the presumptive fin ray regeneration domain. This distal domain was also characterized by expression of ECM remodelers such as *fkbp9*, *mmp11*, *fbln1*, and *emilin1*. In the axolotl limb at 14 dpa, *fkbp9* and *mmp11* were expressed in scattered distal spots, whereas *fbln1* and *emilin1* were enriched around the presumptive chondrogenic front. Together, our findings identified distinct connective tissue cell populations contributing to dermal versus endochondral regeneration in *Polypterus* and highlight conserved and divergent ECM-associated programs between fish fin rays and the tetrapod limb skeleton.

## Reactivation of apical ectodermal ridge-like gene programs in the *Polypterus* wound epidermis

A key driver of successful epimorphic appendage regeneration is the formation of a signaling-competent wound epidermis, which subsequently matures into its specialized derivative, the apical epithelial cap (AEC). To characterize gene expression dynamics associated with wound epidermis formation during *Polypterus* fin regeneration, we subclustered epidermal cell populations corresponding to less differentiated layers (basal and intermediate epidermis) from our snRNA-seq dataset. These included *col17a1*+/*tp63*+ basal cells, *tp63*+ intermediate cells, and *tp63*+/*mki67*+ proliferative epidermal cells (Fig. 4a). At homeostasis (uninjured fin), we detected small populations of basal and proliferative cells, alongside a dominant and well-separated intermediate epidermal cluster. At 1 and 3 dpa, we observed a marked expansion of basal (*col17a1*+) and proliferative (*mki67*+) populations, along with an intermediate cluster whose transcriptional profile shifted toward a basal-like state, consistent with the establishment of a wound epidermis (Fig. 4a, upper panel). Spatial transcriptomics further confirmed epithelial enrichment of the basal cell marker *col17a1* and its pronounced upregulation at 3 dpa (Fig. 4a, lower panel). We observed a difference in the timing of the expression peak for basal epithelial markers such as *col17a1* and others (*cyp27b1* and *krt17*–detailed below) in our snRNA-seq data compared to our spatial transcriptomics data (Fig. 4a, b). Specifically, these markers showed their highest expression levels at 1 dpa in the snRNA-seq and at 3 dpa in the spatial transcriptomics dataset. The intrinsic technical differences between snRNA-seq, which captures newly synthesized nuclear transcripts, and spatial transcriptomics, which primarily reflects accumulated cytoplasmic mRNAs, may explain these differences.

By combining differential gene expression analysis between regenerating and uninjured fins with evaluation of pct.1 (percentage of nuclei expressing a gene within the cluster) and pct.2 (in all other clusters), we identified a set of genes whose expression was highly enriched in the wound epidermis (Supplementary Data 2). Expression of these genes was minimal or undetected in the homeostatic epidermis but became specifically upregulated in discrete subpopulations within the regenerating epidermis, comprising no more than 40% of epidermal cells (Supplementary Fig. 5a, upper panel). A subset of these genes was also expressed in a subpopulation (≤20%) of connective tissue cells (Supplementary Fig. 5a, lower panel). The regenerating epidermis was also marked by the upregulation of several genes with a broader expression pattern (Supplementary Fig. 5b). Inspection of scRNA-seq data from the axolotl limb regeneration and snRNA-seq data from zebrafish caudal fin regeneration (Supplementary Fig. 3a, b) revealed that several *Polypterus* wound epidermis-enriched genes are likewise upregulated in the regenerating epidermis of both axolotl limb and zebrafish caudal fin (Supplementary Fig. 5c–f).

Among these, *krt17*, *cyp27b1*, and *mmp13* showed strong wound epidermis-restricted expression in *Polypterus* (Fig. 4b–d). Using hybridization chain reaction fluorescence in situ hybridization (HCR-FISH) staining, we further validated *krt17* expression predominantly at the basal wound epidermis at 3 and 7 dpa (Fig. 4e). Notably, a marker of activated keratinocytes[23], *Krt17* has been previously reported as enriched in basal wound epidermis cells during axolotl limb regeneration[24]. However, the wound epidermis-restricted expression of *krt17* seems to be specific to *Polypterus*, as in axolotl and zebrafish, expression of the *krt17* orthologs was also detected in the epidermis during homeostasis (Supplementary Fig. 6a, d). *Cyp27b1*, which encodes a key enzyme in vitamin D metabolism[25], has not previously been linked to epimorphic appendage regeneration. In *Polypterus*, its expression was conspicuously enriched in the wound epidermis (barely detected at homeostasis) (Fig. 4c) and may represent an early response to injury (*Cyp27b1* expression peaked at 1 and 3 dpa). The expression of *mmp13*, a gene whose axolotl ortholog has been reported as highly expressed in the limb wound epidermis[26,27], was also highly enriched in the *Polypterus* fin regenerative epidermis but with a later temporal peak of expression compared to *cyp27b1* (Fig. 4d). Both *Cyp27b1* and *Mmp13* displayed regeneration-enriched expression in our axolotl limb spatial transcriptomics series as well as in the axolotl scRNA-seq dataset, however expression was stronger in mesenchymal cells (Supplementary Fig. 6b, c). In zebrafish, *cyp27b1* expression remained consistently low across all stages (Supplementary Fig. 6e), whereas the expression of both *mmp13* paralogs was upregulated in the epidermis, alongside a connective tissue upregulation of *mmp13a* (Supplementary Fig. 6f, g).

Prior studies identified *Frem2* as a wound epidermis marker during axolotl limb regeneration[24,26]. However, this previously reported *Frem2* gene sequence maps to the *Frem3* locus in the most recent axolotl genome release. *Polypterus frem3* was the top upregulated gene in the subclustered epidermal cells at 7 dpa (Supplementary Data 2). We therefore analyzed the expression of both paralogs across *Polypterus*, axolotl, and zebrafish datasets (Supplementary Fig. 7). In both *Polypterus* and axolotl, *frem3*/*Frem3* were expressed in a distinct subset of basal epidermal cells enriched during regeneration (Supplementary Fig. 7a, c). In *Polypterus*, *frem2* was also upregulated in the basal epidermis during regeneration (Supplementary Fig. 7b), but in axolotl, *Frem2* expression was not specifically associated with regenerative stages (Supplementary Fig. 7d). In zebrafish, *frem2a/b* and *frem3* were strongly enriched during regeneration, although their expression was more broadly distributed across basal epidermal and non-basal epidermal cell types (Supplementary Fig. 7e–g).

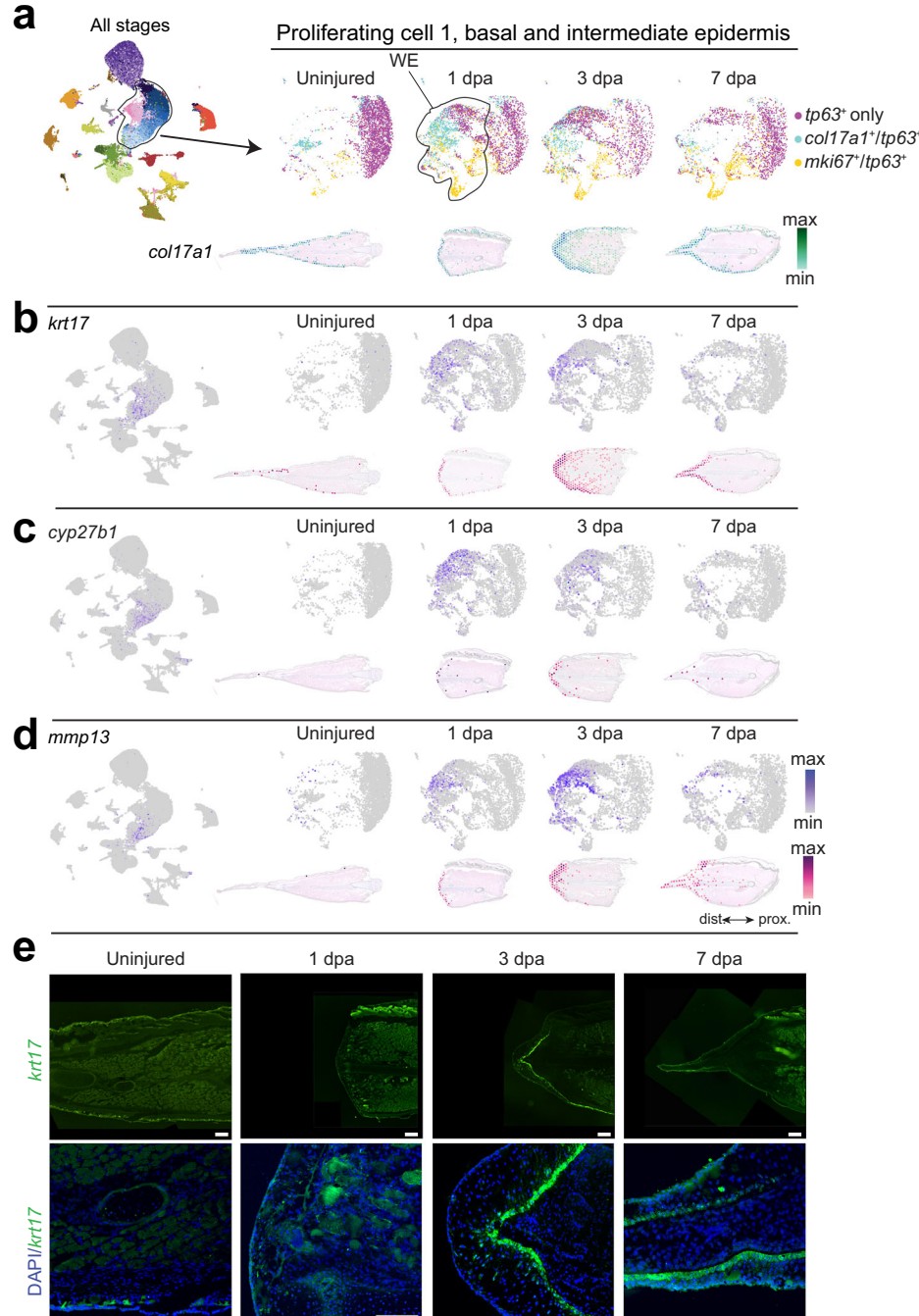

**Fig. 4 | Transcriptional dynamics of wound epidermis formation during *Polypterus* fin regeneration. a** UMAP plots of all stages combined; region outlined demarks the cell clusters used in subclustering analysis of epidermal cell populations; UMAP plots of subclustered *tp63⁺*, *col17a1⁺/tp63⁺*, and *mki67⁺/tp63⁺* cells (top) and spatial expression of *col17a1* (bottom) in the uninjured tissue and throughout 1 dpa, 3 dpa, and 7 dpa stages. UMAP plots and spatial expression of *krt17* (**b**), *cyp27b1* (**c**), *mmp13* (**d**) throughout homeostasis (uninjured) and regeneration stages. **e** HCR-FISH for *krt17* (top) and zoomed in panels of *krt17* counterstained with DAPI (bottom) at homeostasis, 1 dpa, 3 dpa, and 7 dpa. WE, wound epidermis. Scale bars = 100 μm. Proximal (prox.) distal (dist.) axis indicated in (**d**).

The establishment of an AEC is a critical step in epimorphic regeneration, as it defines a specialized signaling center that regulates blastema formation and growth. In amphibians, previous studies have shown that the gene expression program of the apical ectodermal ridge (AER)−essential for limb development−is partially reactivated during AEC formation in regenerating limbs[26]. In *Polypterus* fins at 1 and 3 dpa, we identified a subset of basal epidermal cells with a high score for the expression of AER-related genes, consistent with their involvement in establishing an AEC-like structure during fin regeneration (Fig. 5a). Similar to axolotl[26], part of the AEC transcriptional program in *Polypterus* was also expressed in connective tissue cells (Fig. 5b–i), particularly BMP-related (*bambi*, *bmp2*, *bmp7*) and Wnt-related (*axin2*, *lef1*) genes. Although the epidermal contribution to the AEC program appears greater in *Polypterus* than in axolotl (Fig. 5f–i) or zebrafish (Fig. 5j–m), the participation of connective tissue cells in AEC-associated gene expression seems to be a conserved feature across fish and amphibians.

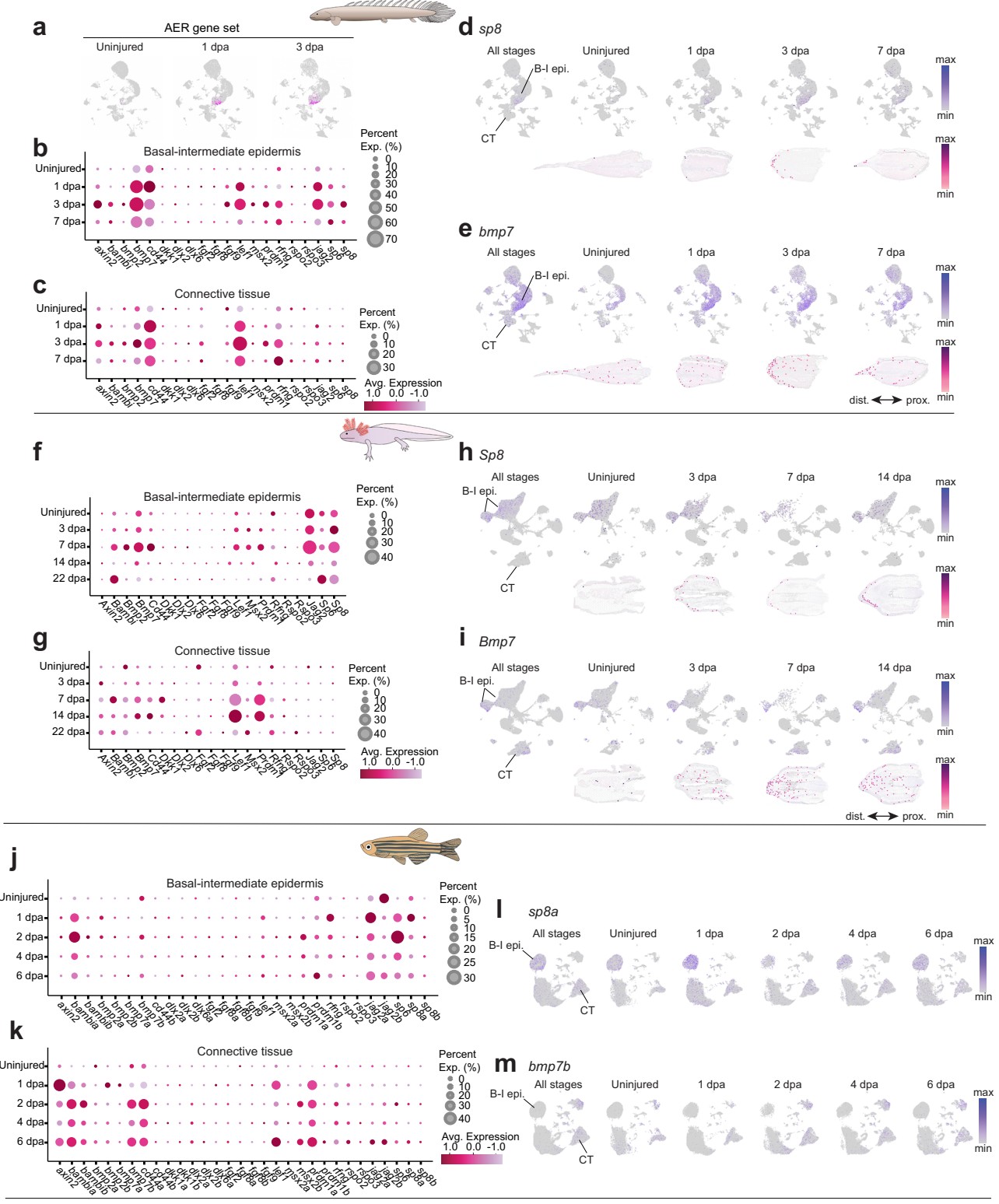

**Fig. 5 | *Polypterus* reactivates AER-related genes during regeneration. a** UMAP plots highlight cells with high expression of an apical-ectodermal ridge (AER) gene set in *Polypterus* fins at homeostasis (uninjured), 1 dpa, and 3 dpa. Dot plots of AER gene expression in the basal-intermediate epidermis (**b**) and connective tissue (**c**) in *Polypterus*; UMAP plots and spatial expression of *sp8* (**d**) and *bmp7* (**e**) in *Polypterus* across uninjured and regenerating stages. **f-i** Dot plots of AER gene expression in the basal-intermediate epidermis (**f**) and connective tissue (**g**) in axolotl; UMAP plots and spatial expression of *Sp8* (**h**) and *Bmp7* (**i**) in axolotl across uninjured and regenerating stages. Dot plots of AER gene expression in the basal-intermediate epidermis (**j**) and connective tissue (**k**) in zebrafish; UMAP plots showing expression of *sp8a* (**l**) and *bmp7b* (**m**) in zebrafish caudal fin across uninjured and regenerating stages. CT, connective tissues; B-I epi., basal-intermediate epidermis. Proximal (prox.) distal (dist.) axis indicated in (**e, i**).

## Blastema cell heterogeneity

Fibroblast-like connective tissue cells are considered the major cellular contributors to the regenerating limb blastema[24,28]. Our snRNA-seq and spatial transcriptomics data identified *col6a3* as a general marker for connective tissue cells (Fig. 2b–d). Connective tissue cells expressing *col6a3* were present during homeostasis and across all stages of regeneration (Fig. 6a). Spatial transcriptomics data showed that during regeneration, *col6a3* expression became more abundant distally, indicating migration of connective tissue cells into the region of the presumptive fin blastema at 3 and 7 dpa (Fig. 2b). To confirm the location of those cells at homeostasis and during regeneration, we used HCR-FISH staining (Fig. 6b). In the uninjured fin, connective tissue cells expressing *col6a3* were found in the dermis and dispersed between muscle cells. At 1 dpa, clusters of *col6a3*-expressing cells were observed in the stump, near the amputation site. At 3 dpa, *col6a3*-expressing cells accumulated distally in the regenerating fin and were abundant in the presumptive blastema. At 7 dpa, *col6a3*-expressing cells were most enriched distally, and *col6a3* signal intensity was higher in cells closest to the epidermis basal layer. Ultimately, *col6a3* expression confirmed that connective tissue cells are major contributors to the *Polypterus* fin blastema.

Regeneration-enriched connective tissue states were characterized by the expansion of small, distinct subpopulations, emphasizing the heterogeneity of the connective tissue compartment during regeneration. For example, *mmp19*, an established marker of axolotl limb blastema cells[29,30], was expressed in a subset of connective tissue cells in the *Polypterus* fin (Fig. 6c). We also observed the expansion of largely non-overlapping connective tissue subpopulations expressing *fgf10* (Fig. 6d), an important signaling factor in limb development and regeneration[31], as well as *lep* (Fig. 6e), whose axolotl ortholog is upregulated in the limb blastema[32].

Although seemingly dispensable for axolotl limb regeneration[33], *Pax7*+ muscle satellite cells contribute to the axolotl limb blastema[34]. In the uninjured *Polypterus* fin, *pax7*-expressing cells were detected in the muscle cluster. Later, at 1 and at 3 dpa, *pax7*-expressing cells were also detected within the connective tissue clusters (Fig. 6f). Myeloid cells, particularly macrophages, infiltrate the limb blastema during regeneration. Our snRNA-seq dataset identified a substantial increase in the proportion of myeloid cells during regeneration, relative to homeostasis (Fig. 1g–j and Supplementary Fig. 2c). To assess whether these cells reached the fin blastema, we examined the expression of *marco*, which marks a subpopulation of myeloid cells in the axolotl limb blastema[24,35]. As seen in the axolotl, our snRNA-seq dataset showed *marco*-expressing cells in the myeloid cluster during *Polypterus* fin regeneration (Fig. 6g, upper panel). Spatial RNA-seq revealed expression of *marco* in the presumptive blastema of the *Polypterus* fin at 3 dpa (Fig. 6g, lower panel).

Altogether, these findings suggest that, as seen in axolotls, the *Polypterus* fin blastema is composed of a heterogeneous cell population, with contributions from tissues such as presumptive muscle satellite cells (*pax7*), connective tissue cells (*col6a3, mmp19, fgf10, lep*), and immune cells (*marco*).

## Emergence of proximal-distal organization within the regenerating connective tissue

While our initial analysis using *K*-means = 4 clustering provided a consistent framework to define the major tissue compartments across all samples in our spatial transcriptomics datasets, we next asked whether finer-grained patterns could be detected within individual compartments. In particular, we noted that at later regenerative stages (*Polypterus* fin at 7 dpa and axolotl limb at 14 dpa), increasing the resolution to *k* = 5 subdivided the connective tissue cluster into two transcriptionally and spatially distinct territories (Fig. 7a, b). One domain localized distally beneath the wound epidermis and was enriched for *crabp2* and *tnc*, while the other occupied more proximal regions and expressed *sfrp2* and *dpt* (Fig. 7c, d). This subdivision was observed in both *Polypterus* fins and axolotl limbs, suggesting that the connective tissue progressively acquires proximal-distal organization as regeneration advances.

To systematically assess the gene expression profile of these domains, we retrieved the top 100 differentially enriched genes in each compartment that could be annotated to a human ortholog (Supplementary Data 3). This orthology-based approach ensured comparability across species and highlighted conserved compartment-specific signatures. Comparison of proximal (*dpt*+, *sfrp2*+) and distal (*tnc*+, *crabp2*+) connective tissue domains revealed both shared and distinct features. Both compartments expressed markers of ECM remodeling, stress responses, and inflammation, consistent with the regenerative wound environment (Fig. 7e and Supplementary Data 3). However, proximal connective tissue showed stronger enrichment for sarcomeric and contractile genes, suggestive of myofibroblast-like states or presence of muscle tissue of the stump. In contrast, distal connective tissue was characterized by high proliferative and secretory activity, reflected in the upregulation of histone and RNA-processing genes, protein-folding machinery, and matrix-associated genes such as fibrillin-2 and thrombospondins.

The identity of these domains was further supported by scRNA-seq and snRNA-seq data (Fig. 7f–k). Subclustering analysis of connective tissue cells indicated that *dpt*+ and *tnc*+ fibroblasts define largely distinct, though partially overlapping populations, suggesting that proximal and distal connective tissue states are transcriptionally and spatially segregated during regeneration. To assess whether these connective tissue domains emerge during zebrafish fin ray regeneration, we evaluated the snRNA-seq dataset of caudal fin ray regeneration. The subclustered zebrafish connective tissue cells showed largely exclusive expression domains of *dpt*+ and *tnca*+/*tncb*+ cells (Fig. 7h, k), with very few connective tissue cells expressing *dpt*. In all three species, *tnc*+ distal connective tissue cells displayed elevated expression of proliferation markers such as *pcna* and *mki67* (Fig. 7l–n), consistent with the higher proliferative activity expected near the wound site. Altogether, our results revealed the emergence of *crabp2*+/*tnc*+ proliferative fibroblasts distally and *sfrp2*+/*dpt*+ fibroblasts proximally during limb and fin regeneration.

## Limb and fin regeneration followed by endoskeleton-level amputation is marked by robust expansion of *hif4a*+ erythrocytes

Our assessment of cellular populations emerging during *Polypterus* fin regeneration revealed a marked increase in erythrocytes after endoskeleton-level fin amputation (Fig. 1g–j). Spatial RNA-seq analysis of the erythrocyte-specific marker *alas2* showed a broad distribution of positive spots during *Polypterus* fin regeneration, with *alas2* expression detected near the wound site and more proximally, in the fin stump (Fig. 8a). To evaluate whether this erythrocyte expansion is a shared feature of limb and fin regeneration we evaluated *Alas2* gene expression during axolotl limb regeneration as well as its zebrafish ortholog during caudal fin ray regeneration. As seen in *Polypterus*, *Alas2*-expressing erythrocytes massively expand during 3 and 7 dpa, near the wound site and in the limb stump, to subside in later stages of axolotl limb regeneration stages (Fig. 8b). In contrast, erythrocyte clusters were not detected during zebrafish caudal fin ray regeneration (Fig. 8c), suggesting that this is a feature of regrowth post endoskeleton-level amputation.

In the Japanese fire-bellied newt (*Cynops pyrrhogaster*), erythrocyte clumps appear during limb regeneration and express genes encoding secreted growth factors and matrix metalloproteases[17]. In contrast, our analysis of the top 100 differentially expressed genes in the erythrocyte cluster versus all other clusters found no strong evidence supporting *Polypterus* or axolotl erythrocytes as a significant source of secreted growth factors or extracellular matrix remodelers

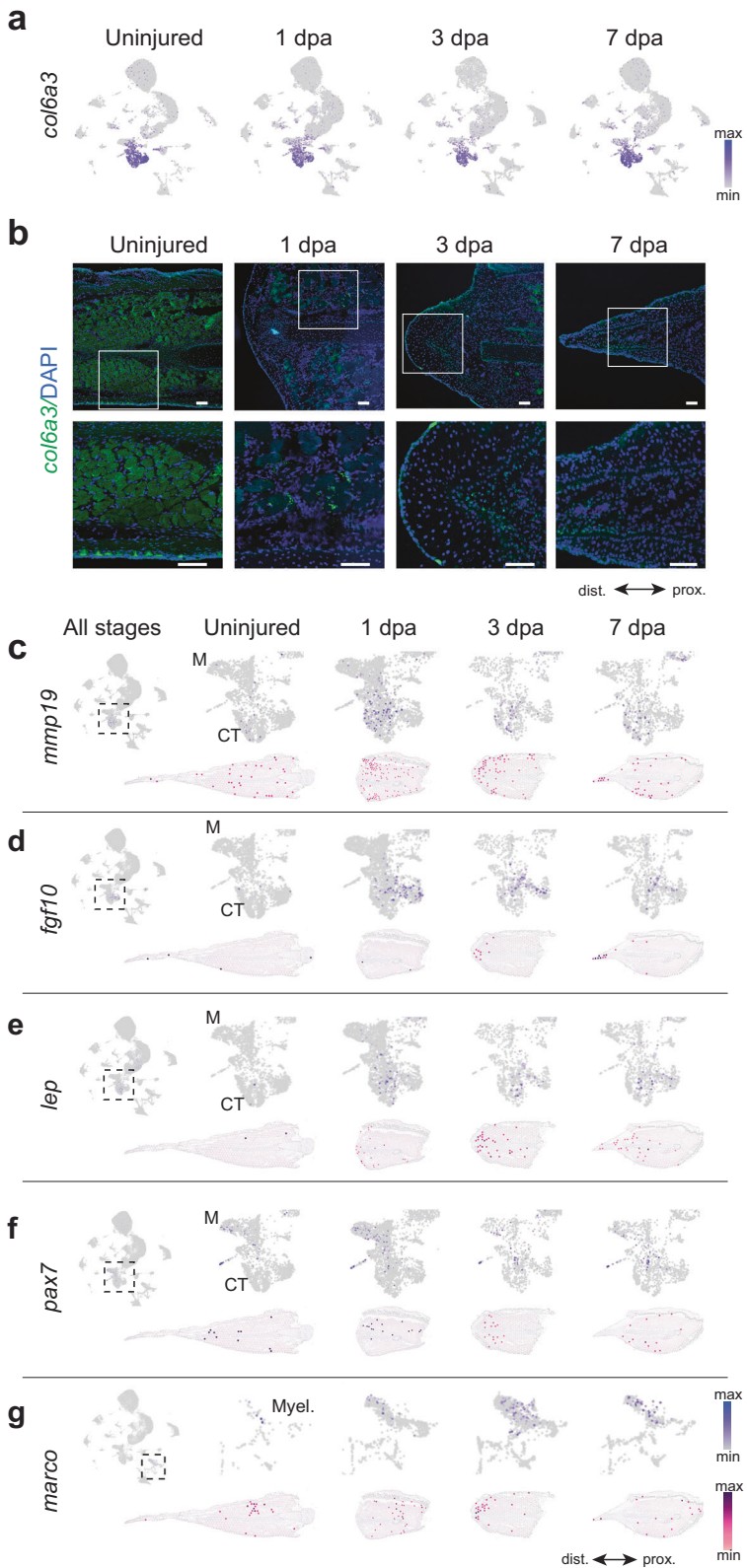

**Fig. 6 | Heterogeneous cell population within the *Polypterus* blastema. a** UMAP plots of cell clusters showing *col6a3* gene expression in the uninjured tissue and across *Polypterus* fin regeneration stages. **b** HCR-FISH panels show *col6a3* (green) and DAPI (blue) staining in uninjured and regenerating *Polypterus* fin tissues; white boxes denote zoomed regions (bottom panels). UMAP plots and spatial expression of *mmp19* (**c**), *fgf10* (**d**), *lep* (**e**), *pax7* (**f**), and *marco* (**g**) in all stages combined and in uninjured and 1 dpa, 3 dpa, and 7 dpa. Box with dashed lines denotes zoomed in regions in subsequent panels; M muscle, CT connective tissue, Myel myeloid cells. Proximal (prox.) distal (dist.) axis indicated in (**b**, **g**). Scale bars = 100 μm.

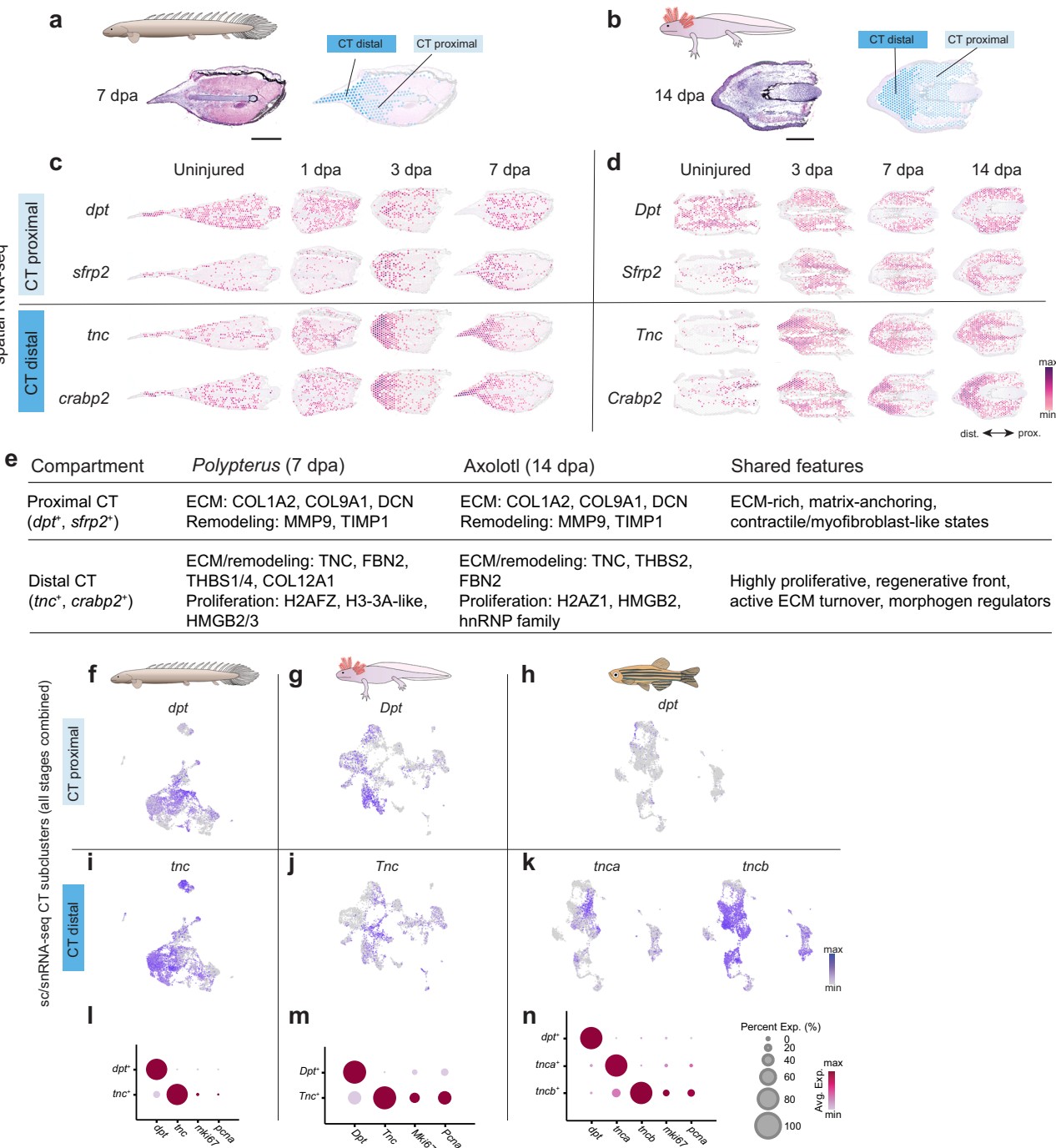

**Fig. 7 | *Polypterus* fin and axolotl limb regenerating connective tissues present transcriptionally distinct proximal and distal territories.** Transcriptionally distinct proximal (light blue) and distal (dark blue) CT domains in both *Polypterus* fin at 7 dpa (**a**) and axolotl limb at 14 dpa (**b**). Spatial expression of *dpt/Dpt* and *sfrp2/Sfrp2* in the proximal CT cluster and of *tnc/Tnc* and *crabp2/Crabp2* in the distal CT of *Polypterus* fin (**c**) and axolotl limb (**d**) in uninjured and regeneration stages. **e** Gene expression summary of proximal and distal CT compartments. UMAP plots of CT subclusters showing *dpt* and tnc expression in *Polypterus* (**f, i**), axolotl (**g, j**), and zebrafish (**h, k**). Dot plot depicting association between *dpt*+ and *tnc*+ cells and the expression of *mki67* and *pcna* in *Polypterus* (**l**), axolotl (**m**), and zebrafish (**n**). CT, connective tissue; ECM, Extracellular Matrix. Proximal (prox.) distal (dist.) axis indicated in (**d**).

(Supplementary Data 4). In adult newts, erythrocyte clumps express *Newtic1*, a salamander-specific gene and putative component of membrane vesicles[36]. However, contrary to adults, *Newtic1*+ erythrocytes in the regenerating limbs of larval newts do not contribute to the blastema, and *Newtic1* expression is limited to gland-like structures along the skin. Likewise, our findings revealed that the axolotl *Newtic1* ortholog was not expressed in the blastema but was instead confined to epidermal cells in both homeostatic and regenerating axolotl limb, as shown by in spatial RNA-seq datasets (Supplementary Fig. 8a). These findings support the hypothesis that *Newtic1*+ erythrocyte clumps are a feature exclusive of post-metamorphic urodele limb regeneration.

Our analysis of the top 100 upregulated genes upregulated in *Polypterus* and axolotl erythrocytes also revealed expression of canonical erythroid markers, such as *rhag2* and hemoglobin b subunits, as well as species-specific upregulation of genes implicated in

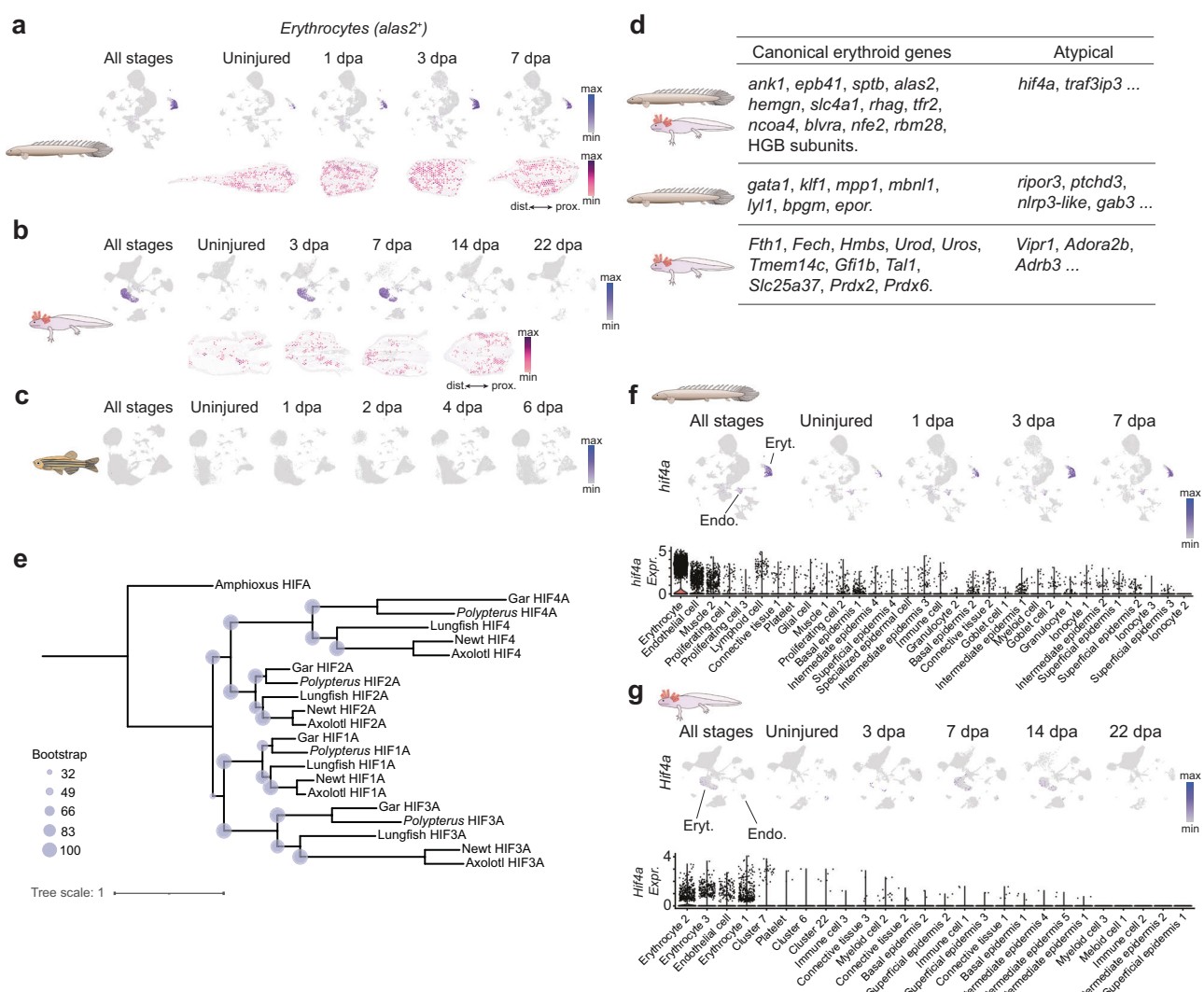

**Fig. 8 | Limb and fin regeneration followed by endoskeleton-level amputation is marked by robust expansion of *hif4a*⁺ erythrocytes.** UMAP plots and spatial RNA-seq showing *alas2* expression in *Polypterus* (**a**) and its axolotl ortholog (**b**) in all stages combined, uninjured appendage, and regeneration stages. **c** UMAP plots showing *alas2* expression in zebrafish caudal fin in all stages combined, uninjured fin, and regeneration stages. **d** Comparison of *Polypterus* and axolotl erythrocyte clusters showing examples of canonical and atypical erythrocyte genes expressed in each species. **e** phylogenetic analysis of HIFA orthologs. **f** UMAP plots (top) of *hif4a* expression in the *Polypterus* fin (**f**) and axolotl limb (**g**) in all stages combined, uninjured, and regeneration stages; Violin plots (bottom) of *hif4a* expression across major cell populations in the *Polypterus* fin (**f**) and axolotl limb (**g**); Eryt, erythrocytes; Endo, endothelial cells. Proximal (prox.) distal (dist.) axis indicated in (**a**, **b**).

modulation of inflammatory response, such as *Vipr1* in axolotl erythrocytes[37], and *gab3* in *Polypterus* erythrocytes[38] (Fig. 8d and Supplementary Fig. 8b, c). Interestingly, both species showed erythrocyte upregulation of *traf3ip3*, a regulator of MAPK signaling and thymocyte development[39], and of a paralog of the hypoxia-inducible factor 1 alpha (*hif1a*) gene, known for its role in translating changes of oxygen levels into cellular responses essential for wound healing and regeneration[40]. Phylogenetic analysis revealed that the *Polypterus* and axolotl *hif1a-like* paralogs upregulated in erythrocytes corresponded to *hif4a*, a homolog to the ancestral vertebrate HIFA gene, which was retained in only some ray-finned fish lineages, including zebrafish[41] (Fig. 8e). In both species, upregulation of *hif4a* occurs almost exclusively in the erythrocyte and endothelial cell clusters (Fig. 8f, g). Finally, *hif4a*-expressing cells were nearly absent from the zebrafish snRNA-seq dataset (Supplementary Fig. 8d). These findings suggest that the emergence of *hif4a*⁺ erythrocytes is unique to limb and fin regeneration after endoskeleton-level amputation and the role of erythrocytes in this context may extend beyond oxygen transport.

## Hypoxia response mechanisms during *Polypterus* fin regeneration

Evidence from various regenerative models, including mouse[42], *Xenopus*[43], axolotl[44], and zebrafish[45], supports a scenario where wounding drives local reactive oxygen species (ROS) production, which in turn promotes a hypoxic environment permissive to regeneration. HIF1A plays a crucial role in translating changes in oxygen levels into cellular responses essential for wound healing and regeneration[40]. HIF1A levels are tightly regulated by prolyl hydroxylase domain (PHD) enzymes (encoded by the *EGLN1*, *EGLN2*, and *EGLN3* genes) and by the von Hippel-Lindau protein (pVHL, encoded by the *VHL* gene), which promote HIF1A degradation, and HIF1AN (encoded by the *HIF1AN* gene), which negatively regulates *HIF1A* transcription. Hypoxia prevents degradation of HIF1A by its negative regulators, resulting in HIF1A-induced increase in the expression of glycolytic enzymes[46,47].

Recent findings demonstrated that during *Xenopus* and axolotl limb regeneration, the expression of HIF1A negative regulators is low,

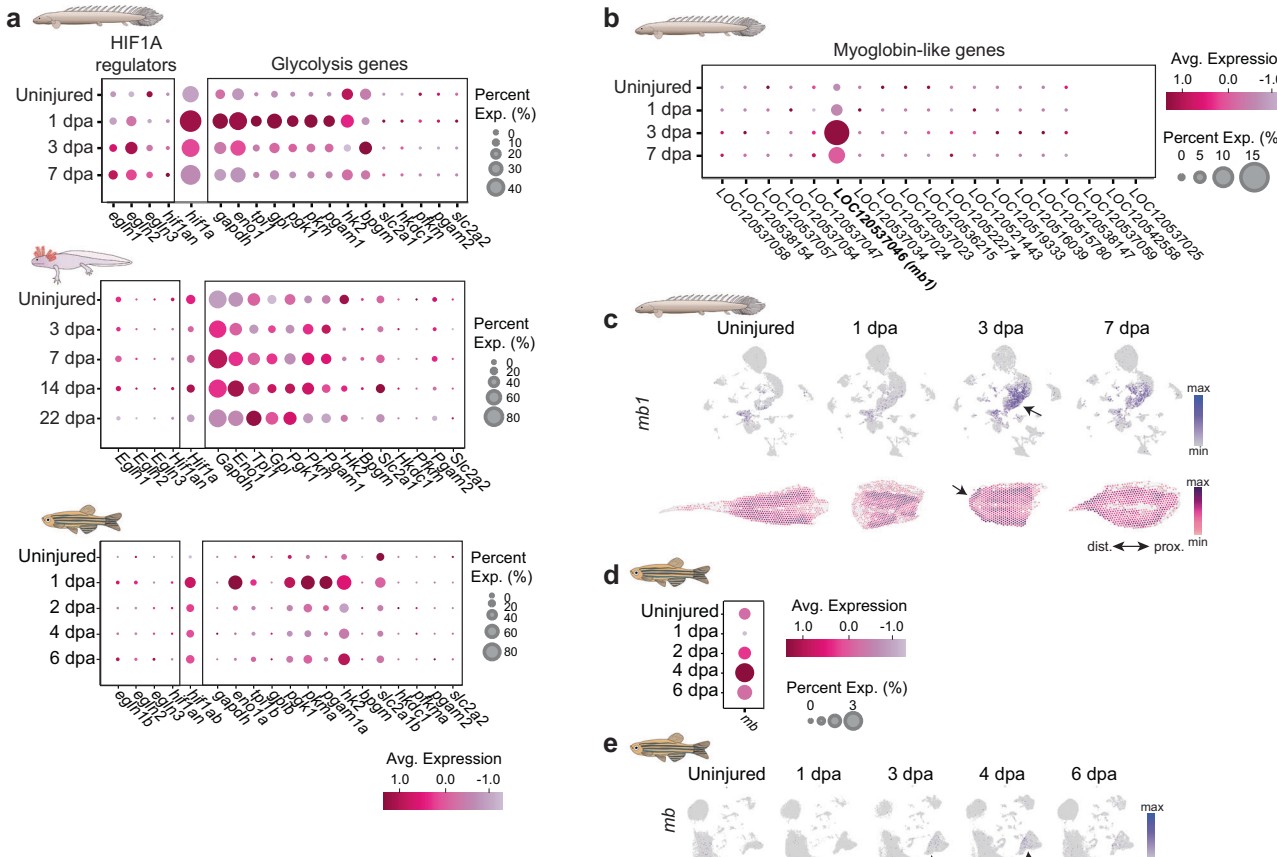

**Fig. 9 | Hypoxia responses during *Polypterus* fin regeneration involve the regulation of HIF1A, glycolysis, and upregulation of myoglobin gene expression.** **a** Dot plots showing the expression of HIF1A regulators and glycolysis genes during regeneration of the *Polypterus* fin (upper panel), axolotl limb (middle panel), and zebrafish caudal fin (lower panel). **b** Dot plot showing the expression of myoglobin-like genes during *Polypterus* fin regeneration. **c** UMAP plots (top) and spatial RNA-seq (bottom) showing expression of *mb1* in the uninjured *Polypterus* fin and during regeneration; arrows indicate expression in epidermal cell clusters (top) and wound epidermis domain (bottom). **d** Dot plot of *mb* expression during zebrafish caudal fin regeneration. **e** UMAP plots showing *mb* expression in the uninjured zebrafish caudal fin and during regenerating stages; arrows indicate expression in the basal epidermis clusters. Proximal (prox.) distal (dist.) axis indicated in (**c**).

while *Hif1a* gene and glycolysis-related genes are highly expressed[48]. This suggests that these species stabilize HIF1A by maintaining low expression levels of its regulators during regeneration. We examined the expression of the *Polypterus hif1a* and its regulators, as well as glycolysis genes, in our spatial and snRNA-seq datasets. As seen in *Xenopus* and axolotl, *Polypterus egln1, egln2, egln3*, and *hif1an* expression levels remained mostly low at both uninjured and 1 dpa stages, whereas *hif1a* expression was greatly increased (Fig. 9a, upper panel). At 3 dpa, even as expression of HIF1A regulators increased, and *hif1a* expression decreased, *hif1a* remained broadly expressed in the blastema region, in contrast to its regulators (Supplementary Fig. 9a, b). Concordantly, expression dynamics of glycolysis genes largely mirrored that of *hif1a* (Fig. 9a, upper panel), and most were broadly expressed in the *Polypterus* fin at 3 dpa, especially in the muscle (Supplementary Fig. 9c). We further confirmed that during axolotl limb (Fig. 9a, middle) and zebrafish caudal fin ray (Fig. 9a, lower panel) regeneration, a comparable upregulation of /HIF1a/hif1ab and glycolysis genes occurred, although in the zebrafish this program was mostly limited to the 1 dpa stage. Altogether, these findings suggest an evolutionarily conserved regenerative response to hypoxia, involving HIF1A expression stabilization and activation of glycolysis.

Myoglobin (MB) is a versatile protein shown to mitigate both ROS activity and hypoxia. Specifically, high MB levels in muscle cells can increase oxygen storage capacity in marine mammals during extended dives[49]. Furthermore, MB attenuates oxidative stress in cardiac muscle[50]. Remarkably, while most vertebrates possess a single MB gene, *Polypterus* possesses at least 15 *mb* genes[51], whereas frogs and salamanders have lost their MB orthologs[52]. We examined the expression of *Polypterus mb* genes in our snRNA-seq dataset and found 19 genes currently annotated as myoglobin-like genes in the *Polypterus* genome. Of those, *LOC120537046* was markedly upregulated in the wound epidermis during fin regeneration (Fig. 9b). A previous study showed that among *Polypterus* myoglobin-like proteins, PseMb1−encoded by *LOC120537046*−exhibited the highest sequence identity to the mammalian *MB* gene[51]. Here, we refer *LOC120537046* as *mb1*. In the uninjured and 1 dpa fin tissue, *mb1* was expressed in the muscle and in a few cells in the basal epidermis clusters (Fig. 9c, upper panel). At 3 dpa, *mb1* expression in the basal epidermis increased, and then moderately decreased at 7 dpa. Spatial RNA-seq showed *mb1* broadly expressed at 3 dpa, with the highest levels in the muscle and wound epidermis (Fig. 9c, lower panel). Interestingly, the zebrafish *mb* gene was also upregulated during caudal fin ray regeneration, peaking at 4 dpa (Fig. 9d), and its expression was mostly detected in basal epidermis cells at 3 and 4 dpa (Fig. 9e). While the roles of the *Polypterus mb1* gene in fin regeneration remain unclear, our findings suggest that *mb* and its paralogs may be deployed as components of a general mechanism for coping with oxygen stress during fin regeneration, providing a foundation for further investigation into their functional significance.

## Activation of DNA damage and repair and switch from pro- to anti-inflammatory immune response during limb and fin regeneration

Previous studies on diverse regenerative models, including planarians[53], newts[54], *Polypterus*[4], and axolotl[4], have demonstrated the upregulation of DNA damage and repair genes at the onset of regeneration. Activation of DNA damage response (DDR) has been shown to be required for regeneration of both the axolotl limb[55] and the zebrafish heart[56]. Analysis of our snRNA-seq data revealed upregulation of DDR markers during fin regeneration. For *Polypterus* and zebrafish, fin amputation resulted in upregulation of DNA damage sensing (*atm*, *atr*, *tp53*) and repair (*rad51*, *rpa2*, *fen1*, *chaf1a/chaf1b*) markers, with DNA damage sensing genes being observed in a larger number of cells when compared to DNA damage repair markers (Supplementary Fig. 10a, b). In the axolotl, DNA damage sensing and repair genes were expressed throughout regeneration in relatively similar proportions of cells (Supplementary Fig. 10c). Furthermore, in the axolotl scRNA-seq dataset, the expression of DNA damage sensing and repair genes was already detected at homeostasis. Spatial transcriptomics showed DNA damage sensing and repair gene expression becoming progressively distalized in both *Polypterus* fin and axolotl limb during regeneration (Supplementary Fig. 10d, e). These findings corroborate a common activation of DDR during fin and limb regeneration of the *Polypterus* fin and the axolotl limb.

Following axolotl limb amputation, pro- and anti-inflammatory signals are induced and sustained through regeneration[16], with pro-inflammatory macrophages peaking during early to mid-blastema stages and anti-inflammatory macrophages being predominant at later stages[57]. Likewise, in larval zebrafish, caudal fin amputation results in accumulation of pro-inflammatory macrophages at the wound site at 1 dpa, followed by an increase in anti-inflammatory macrophages[58]. Our analysis of the immune response landscape of *Polypterus* fin regeneration broadly parallels the pattern described in both the axolotl and zebrafish. Upon fin amputation, at 1 dpa, we observed upregulation *of il1b* expression, a classic marker of pro-inflammatory macrophages[59], in a subset of myeloid cells (Supplementary Fig. 11a, b). At 1 dpa, the myeloid cluster also expressed high levels of *spp1*, a fibrogenic macrophage marker[60], which was sharply downregulated at subsequent stages. Other candidate orthologs of mammalian genes associated with pro-inflammatory macrophages, such as *tnfsf15*[61] and *hsp90b1*[62], were upregulated at 1 dpa. Anti-inflammatory markers were also upregulated at 1 dpa, but their expression peaked at subsequent regeneration stages (Supplementary Fig. 11a, c). *tgfb1*, an established immune modulator frequently presenting anti-inflammatory activity, was gradually upregulated (Supplementary Fig. 11a) and broadly expressed in cells of the myeloid cluster (Supplementary Fig. 11c) during regeneration stages. Expression of other putative orthologs of anti-inflammatory markers, *hs3st1-like*, *clec7a-like*, and *mcf2l*, also peaked at later stages of fin regeneration. While some overlap exists between pro- and anti-inflammatory marker expression at 1 dpa, differences in the subset of cells expressing *il1b/spp1* and *hs3st1l1/clec7a-like* within the myeloid cluster (Supplementary Fig. 11b, c) suggest that pro- and anti-inflammatory macrophages coexist in early regeneration stages. Most pro- and anti-inflammatory markers described here were expressed below the sensitivity level of spatial transcriptomics. However, we did detect expression of *spp1* (Supplementary Fig. 11d) and *clec7a-like* (Supplementary Fig. 11e) in our *Polypterus* spatial RNA-seq dataset. At 1 dpa, *spp1* was readily detected in the wound epidermis and distal mesenchyme, whereas *clec7a-like*⁺ spots were found in the fin stump. At 3 dpa, positive spots for both genes were detected distally, within and around the wound epidermis and subjacent blastema. Although detection of *spp1* and *clec17a-like* was limited to a few dozen spots, the early detection of *spp1* near the injury site at 1 dpa, but not *clec7a-like*, is consistent with the initial wave of pro-inflammatory signals we detected in our snRNA-seq dataset. Altogether, these data

support a similar dynamic of pro- and anti-inflammatory immune cell response during fin and limb regeneration.

## Epigenetic analysis uncovers candidate fin regeneration-responsive elements

The identification of Tissue Regeneration Enhancer Elements (TREEs)[63] active during limb and fin regeneration may ultimately reveal conserved gene regulatory networks that orchestrate a successful regenerative program. Various TREEs active during teleost fish fin ray regeneration have been identified in recent years[8,64–66], yet changes to the epigenetic landscape in response to injury at the fin endoskeleton region—the homologous counterpart to the tetrapod limb—remain entirely unexplored. To address this, we used ATAC-seq to assess chromatin accessibility in the uninjured fin and at 3 dpa—a stage in which a signaling-competent wound epidermis is established, and blastemal cells are readily detected. Biological triplicates of uninjured and 3 dpa fin samples were used for ATAC-seq library preparation. Over 150 million paired-end reads were sequenced per sample, and the resulting libraries showed the expected fragment periodicity, separation between conditions, and clustering between replicates, along with over thirty-two and forty-seven thousand peaks called for the uninjured and 3 dpa conditions, respectively (Supplementary Fig. 12a–c and Supplementary Data 5). Our analysis revealed 1,427 differentially accessible chromatin regions (adjusted *p*-value < 0.05), or peaks, between the 3 dpa and the uninjured fin tissue (Fig. 10a and Supplementary Data 6). Of those, 155 peaks mapped to unplaced DNA scaffolds. The 1272 peaks that mapped to *Polypterus* chromosomes were mostly (>86%) intergenic (Supplementary Fig. 12d).

Focusing on the top 100 differentially accessible peaks ranked by statistical significance, we searched for potential gene targets, reasoning that genes within their regulatory range would be differentially regulated during fin regeneration. To this end, we generated bulk RNA-seq data from uninjured and 3 dpa *Polypterus* fins in biological triplicates. Over 40 million paired-end reads were sequenced per sample (Supplementary Data 7), and resulting libraries showed separation between conditions and clustering of replicates (Supplementary Fig. 12e). We found 4393 differentially expressed genes (Fig. 10b and Supplementary Data 8). Next, we searched for differentially expressed genes (fold change > 2, adjusted *p*-value < 0.05) within 600 kb up or downstream of our top 100 intergenic peaks. For peaks located in a gene desert, we expanded the search to 1.2 mb up or downstream of the peak. We found 45 genes differentially expressed at 3 dpa located in the vicinity of our top 100 ATAC-seq peaks (Fig. 10b). This list included genes such as *lep* (Fig. 10a, b, Supplementary Fig. 12f–h), highlighted as a marker for a subpopulation of regeneration associated connective tissue cells (Fig. 6e), and *bmp7* (Fig. 10a, b, Supplementary Fig. 12i–k), part of the *Polypterus* AER program gene set found upregulated in the regenerating epidermis and connective tissue compartments (Fig. 5b, c, e). In addition, this list included *trim71*, recently shown to potentiate proliferation of reprogrammed limb progenitors[67] and *tes*, identified in our *Polypterus* fin snRNA-seq dataset as upregulated in the regenerating epidermis (Supplementary Data 2). Among the peaks differentially accessible in the uninjured fin, we found peak 66, located near *ccn2b*, a gene highly expressed at homeostasis and downregulated over 6-fold at 3 dpa (Supplementary Fig. 12l–n and Supplementary Data 8). Its downregulation at 3 dpa is consistent with its reported role in promoting fibrosis[68].

Next, we inspected more closely three peaks ranked among the top 20 most significantly changed from the uninjured to the 3 dpa fin stage: peak 5 (within *dip2b* gene and upstream of *lima1*), peak 9 (upstream of *trim71*), and peak 77 (upstream of *tes*). Peak 5, located in intron 22 of *dip2b*, had two genes upregulated in its vicinity: *dip2b* and *lima1* (Fig. 10c). While further experiments will be necessary to confirm the gene under peak 5 influence, expression of *lima1* seems more associated with progression of regeneration. Both genes were

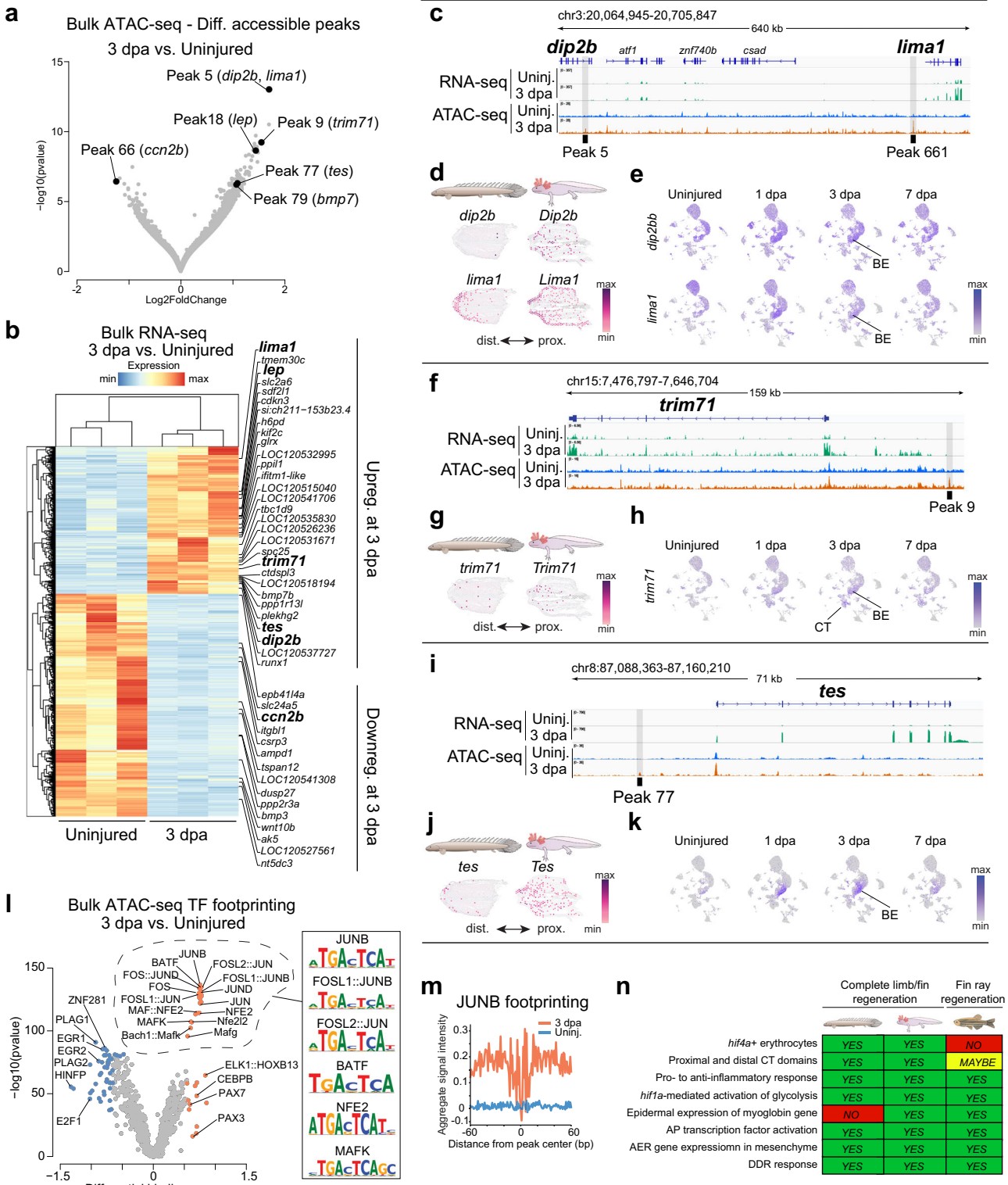

**Fig. 10 | Epigenetic profiling of *Polypterus* fin regeneration uncovers candidate TREEs and enrichment of AP-1 TF footprint.** Differentially accessible peaks (**a**) and differentially expressed genes (**b**) during *Polypterus* fin regeneration. Genomic location of peaks 5 and 661 near *dip2b* and *lima1* (**c**), peak 9 near *trim71* (**f**), and peak 77 near *tes* (**i**). Spatial expression of *Polypterus dip2b* and *lima1* (**d**), *trim71* (**g**), *tes* (**j**), and axolotl orthologs at 3 dpa. UMAP plots of *dip2b* and *lima1* (**e**), *trim71* (**h**), and *tes* (**k**) expression in uninjured and regeneration stages. **l** Volcano plot of TF

differential binding scores; dashed line denotes AP-1 TF family members and predicted binding sites. **m** Aggregate plot of JUNB footprinting. **n** Shared and derived hallmarks of appendage regeneration. BE basal epidermis, CT connective tissue. ATAC-seq fold changes and *p*-values calculated using DiffBind (DESeq2 function); differential binding scores and *p*-values calculated using TOBIAS (BIN-Detect function).

upregulated by just over 2-fold, bulk RNA-seq showed that *lima1* expression at 3 dpa is over 15 times greater than that of *dip2b* (Supplementary Data 8). Of note, this genomic landscape also included peak 661, located 19 kb upstream of *lima1*. Consistent with the bulk RNA-seq expression levels, spatial RNA-seq detected only a few positive spots of *dip2b* expression in the *Polypterus* fin stump at 3 dpa (Fig. 10d). Its axolotl ortholog, *Dip2b*, was detected in scattered spots in the epidermis and mesenchyme (Fig. 10d). SnRNA-seq showed increased *dip2b* expression across all epidermal cell clusters (Fig. 10e). Unlike *dip2b*, *lima1* and its axolotl ortholog *Lima1* were expressed throughout the epidermis, but noticeably highly expressed in the wound epidermis in the spatial transcriptomics data (Fig. 10d). Concordantly, snRNA-seq showed robust *lima1* expression in basal epidermis, with increasing levels of expression as regeneration progressed (Fig. 10e). Peak 9 was located 50 kb upstream of *trim71* (Fig. 10f), which was upregulated at 3 dpa in our bulk RNA-seq dataset. Spatial RNA-seq showed *trim71* expressed in scattered spots along the proximal and distal mesenchyme, and wound epidermis of both the *Polypterus* fin and the axolotl limb at 3 dpa (Fig. 10g). SnRNA-seq showed increased *trim71* expression during fin regeneration, particularly in the connective tissue and basal epidermis clusters (Fig. 10h). Peak 77 was located approximately 13 kb upstream of *tes* (Fig. 10i). Spatial RNA-seq showed *tes* mainly expressed in the *Polypterus* fin wound epidermis at 3 dpa and expressed in scattered spots along the wound epidermis, proximal and distal mesenchyme of the axolotl limb at 3 dpa (Fig. 10j). SnRNA-seq showed expression of *tes* in the basal epidermis during fin regeneration (Fig. 10k).

Next, we investigated genome-wide transcription factor (TF) binding dynamics using TOBIAS[69]. We analyzed differential TF occupancy, or footprinting, across the 1,427 differentially accessible peaks identified between 3 dpa and uninjured fin tissue. The most striking trend in our differential TF binding analysis was the enrichment of bound sites for the AP-1 family of TFs at 3 dpa (Fig. 10l and Supplementary Data 9). Among AP-1 TFs, JUNB exhibited the most statistically significant differential binding at 3 dpa compared to uninjured fin tissue. Consistently, the aggregated JUNB footprint plot for all 1,427 peaks displayed a well-defined JUNB footprint signal in differentially accessible genomic regions at 3 dpa relative to the uninjured fin (Fig. 10m). We examined the expression of *Polypterus* AP-1 TFs in bulk RNA-seq data from 3 dpa versus uninjured fins and found the *Polypterus junb* significantly upregulated at 3 dpa (fold change = 3.3, adjusted *p*-value = 2.25E-12) (Supplementary Fig. 12o, Supplementary Data 9). Spatial transcriptomics data showed *Polypterus junb* and axolotl *Junb* were widely expressed in epidermal and mesenchymal cells, with *Polypterus* showing a more distally concentrated expression of *junb* (Supplementary Fig. 12p). Overall, our findings indicate that, like axolotls and zebrafish, AP-1 TFs play a crucial role in regulating the activity candidate TREEs during *Polypterus* fin regeneration.

## Discussion

Highly regenerative vertebrates may achieve complete limb or fin regeneration by employing conserved genetic and cellular programs inherited from their last common ancestor. At the same time, they may also activate species-specific, derived programs that evolved later and may rely on novel genes. While species-specific programs might be confined to the model organism being studied, ancestral regenerative programs are expected to be shared across multiple regenerative species. Therefore, comparative studies of limb and fin regeneration are essential for distinguishing between ancestral and derived mechanisms, ultimately accelerating the discovery of the core genetic and cellular processes that drive successful regeneration. By utilizing *Polypterus* as a model system, we expand upon insights gained from teleost models such as zebrafish, extending our understanding of fin regeneration to include the endoskeletal region—the homologous counterpart to the tetrapod limb. This region adds tissue and cellular

complexity, including endochondral elements, muscle, nerves, blood vessels, and connective tissue associated to these elements, providing a closer parallel to limb regeneration.

Our study presents a comprehensive cellular and transcriptional analysis of vertebrate appendage regeneration by generating extensive snRNA-seq profiles of *Polypterus* uninjured and regenerating fins, along with spatial RNA-seq datasets from *Polypterus* fins and axolotl limbs at homeostasis and regeneration. We further incorporated published datasets from axolotl and zebrafish appendages, enabling a cross-species comparison at single-cell resolution. This analysis identified both shared and lineage-specific cellular and molecular mechanisms underlying vertebrate appendage regeneration (Fig. 10n).

Our findings revealed the emergence of *crabp2*[+]/*tnc*[+] proliferative fibroblasts distally and *sfrp2*[+]/*dpt*[+] fibroblasts proximally during limb and fin regeneration. Proximal *dpt*[+] fibroblasts expressed *sfrp2*, a secreted Wnt antagonist[70], while distal *tnc*[+] fibroblasts expressed *crabp2*, a regulator of retinoic acid signaling[71,72]. These reciprocal gene expression patterns align with classical morphogen pathways that contribute to the proximal-distal patterning of the connective tissue. Retinoic acid is a well-known proximalizing cue in axolotl limb regeneration[73,74], while Wnt/β-catenin signaling promotes distal identity in developmental contexts, with *sfrp2* acting as a potential proximal stabilizer[75,76]. Our assessment of zebrafish caudal fin regeneration snRNA-seq dataset further revealed distinct connective tissue cells expressing *tnca*[+]/*tncb*[+] and *dpt*[+] markers. Although spatial expression data are unavailable for zebrafish, this pattern suggests that dpt[+]/tnc[+]-based proximal-distal regionalization may also occur during zebrafish caudal fin regeneration.

In both the axolotl limb and zebrafish fin, amputation initiates a sequential activation of pro- and anti-inflammatory programs[16,57,58,77]. Similarly, our snRNA-seq analysis of the *Polypterus* myeloid cluster indicates that following fin amputation, both pro- and anti-inflammatory signals contribute to the wound healing response. As regeneration progresses, anti-inflammatory signals become predominant (Supplementary Fig. 11). Altogether, our findings provide further support for this wound healing dynamics as an evolutionarily shared feature of appendage regeneration[77].

Activation of DNA damage response is a hallmark of limb[55] and fin[78] regeneration. Our analysis showed that this is also true during *Polypterus* fin regeneration (Supplementary Fig. 9). We found that all three models exhibit a comparable upregulation of genes involved in DNA damage sensing and repair upon injury (Supplementary Fig. 9), although the magnitude of this response varies among species. In particular, the DNA damage response was more broadly activated in axolotl, with DDR markers expressed in 30–40% of all cells. In our fish models, fewer than 20% of cells showed expression of DDR genes (Supplementary Fig. 9a–c). Previous findings showed how axolotl limb amputation leads to systemic cell cycle activation throughout the body[79]. Further studies in *Polypterus* and other regenerative species will help establish whether a systemic activation of cell cycle, and potentially DDR, is unique to axolotls or is an evolutionarily shared feature of limb/fin regeneration-competent species.

Through our comparative analysis we identified both shared and species-specific refined features of the wound epidermis transcriptional program. One common feature of fish fin and axolotl limb regeneration was the shared reactivation of AER-related genes by the wound epidermis and connective tissue cells. Previous studies demonstrated that typical genetic programs of the AER—the distal signaling epithelium that helps drive limb outgrowth during embryogenesis—are deployed instead by mesenchymal cells in the axolotl limb during development[80] and regeneration[81]. More recently, comparative analysis of *Xenopus* tadpoles and axolotls showed that transcriptional programs of the *Xenopus* limb wound epidermis were activated in both the wound epidermis and subjacent mesenchyme during axolotl limb regeneration[26]. Consistent with these observations,

we found upregulation of AER-related genes in both the wound epidermis as well as subsets of connective tissue cells in *Polypterus* and zebrafish (Fig. 5). While the overall pattern of AER gene reactivation is conserved across all three models, specific genes showed lineage-specific expression patterns. For instance, epithelial expression of *bmp7* was more robust and widespread in *Polypterus* than in zebrafish or axolotl. We also identified several other genes commonly upregulated in the regenerating epidermis across species (Supplementary Fig. 5). For this shared gene set, however, both the level of upregulation and the degree of epidermal versus mesenchymal expression varied among species. Together, these findings suggest that *Polypterus* regeneration relies on a hybrid epithelial program that integrates ancestral wound epidermis/connective tissue AER reactivation with lineage-specific responses. This integrated program provides new insight into the cellular and molecular origins and evolution of vertebrate regenerative competence.

Our study supports the activation of glycolysis as universal feature of limb and fin regeneration. Urodele amphibians such as newts are capable of regenerating severed limbs, even under high environmental oxygen concentrations[82]. Recently, it has been proposed that regenerative models such as the frog tadpole and the axolotl maintain low expression levels of HIF1A regulators during regeneration[48]. This would enable stabilization of HIF1A and activation of glycolytic genes regardless of oxygen levels. Our results suggest that a similar mechanism may be taking place during *Polypterus* and zebrafish caudal fin regeneration (Fig. 9a). Although the timing and expression dynamics of HIF1A and its regulatory genes vary slightly across species, the upregulation of several glycolysis-related genes during regeneration was consistently observed in all models. A metabolic switch to glycolysis has been implicated in a series of biological processes potentially relevant to regeneration, including maintenance of stemness[83], cell proliferation[84], and modulation of immune cell function[85]. Our findings suggest that a reduced oxygen sensitivity via HIF1A stabilization may be a vertebrate ancestral trait linked to enhanced regenerating capacity.

While activation of a glycolytic pathway appears to be conserved across fin and limb regeneration, another potential adaptation to low oxygen conditions—the epithelial expression of myoglobin—seems to be restricted to fin regeneration. *Polypterus* experienced a remarkable expansion in their myoglobin gene repertoire[51], whereas salamanders and frogs lost their *mb* gene ortholog[52]. We identified a *Polypterus* myoglobin-like gene, termed *mb1*, as highly expressed in the muscle but, surprisingly, also in the wound epidermis. Likewise, albeit to a lesser extent, the zebrafish *mb* gene was also upregulated in the basal epidermis during fin regeneration. Myoglobin gene expression in non-muscle tissues is rare but has been reported in the liver, gills and brain of the common carp (*Cyprinius carpio*) as a potential adaptation to low oxygen environments[86], in human carcinoma cells, where it may help in coping with the hypoxic conditions associated with neoplastic growth[87], and in human hematopoietic progenitors that reside in the hypoxic bone marrow microenvironment[88]. While the role of *mb1* in the *Polypterus* fin wound epidermis is unclear, our findings identify species-specific mechanisms that may be in place to manage oxygen and metabolic demands of regeneration.

Across all three models, regeneration was characterized by an expansion of less differentiated epithelial populations, proliferating cells, and immune cell types, particularly myeloid cells. However, a pronounced influx of erythrocytes at the injury site was not universal. The striking enrichment of *alas2*+ erythrocytes in *Polypterus* and axolotl, but not in zebrafish, and prior observations of erythrocyte clump formation during newt limb regeneration, suggest that erythrocyte recruitment may be a feature of a regeneration program deployed in limbs and in the endoskeleton-bearing region of the fin. We also identified both shared and species-specific features of the regeneration-associated erythrocyte populations (Fig. 8d). In both

*Polypterus* and axolotl, erythrocytes upregulate non-canonical erythroid genes such as *traf3ip3*/*Traf3ip3*, an immune modulator[89], and *hif4a*/*Hif4a*, an ohnolog[90] of *hif1a* that arose from the two rounds of whole-genome duplication at the base of vertebrate evolution. Moreover, erythrocytes in these two species likely do not serve as a major source of secreted growth factors or extracellular matrix remodelers, as proposed for erythrocyte clumps in the newt. Beyond oxygen delivery, erythrocytes may influence regeneration by modulating hypoxia responses, heme metabolism, angiogenesis, or inflammatory signaling within the blastema[91], highlighting their potential metabolic and signaling roles during the evolution of vertebrate regeneration.

Finally, we characterized the gene regulatory landscape underlying *Polypterus* fin regeneration. When integrated with published data from zebrafish and axolotl, our results revealed conserved gene regulatory features across vertebrate appendage regeneration, notably the transcriptional control by AP family transcription factors. The identification of TREEs active during limb and fin regeneration can provide valuable insights into the evolutionary dynamics of gene regulatory networks governing regeneration. As a first step towards this goal, our chromatin accessibility profiling of uninjured and regenerating fins at 3 dpa uncovered hundreds of differentially accessible peaks. Notably, many of the most significant peaks were located near genes that were upregulated during *Polypterus* fin regeneration, such as *lima1*, *trim71*, and *tes* (Fig. 10). Like *tes*, the mammalian *Lima1* has been proposed to act as a mechanosensitive regulator of cell adhesion[92], and both *lima1* and *tes* were upregulated in the wound epidermis of the *Polypterus* fin and axolotl limb. Recently, *Trim71*, along with *Prdm16*, *Zbtb16*, and *Lin28*, has been identified as part of a core set of four factors capable of successfully reprogramming mammalian non-limb fibroblasts into limb progenitor-like cells[67]. Peak 79, also among our top 100 most significant differentially accessible peaks, was found upstream of *bmp7*, a gene implicated in limb regeneration[93]. Moreover, peak 18 was located upstream of *lep*, the *Polypterus* ortholog to *lepb*, a gene sharply upregulated upon zebrafish fin ray amputation[63]. Expression of *lepb* during fin ray regeneration is regulated by a well-characterized TREE, termed *LEN*, located between *lebp* and *mir129-1*[63]. Interestingly, peak 18 was located over 1 mb upstream of the *Polypterus lep* promoter, in a gene desert spanning nearly 3 mb between *mir129-1* and *snd1*. Hence, peak 18 is likely not an ortholog of LEN, but possibly an additional TREE, active in the context of regeneration following fin amputation at the endoskeleton level. Via differential TF footprinting, we identified an enrichment for predicted bound sites of AP-1 TFs during *Polypterus* fin regeneration, similar to recent reports in zebrafish[8,64] and the axolotl[94]. Pending functional validation, the candidate TREEs identified here will represent a valuable resource to studies aimed at identifying shared and derived TREEs controlling limb and fin regeneration. A recent high-quality epigenetic profiling study of axolotl limb regeneration successfully identified and functionally validated an axolotl TREE[94]. However, since sampling consisted of fluorescence-activated cell sorting (FACS) of connective tissue cells, the axolotl dataset lacks the cellular complexity present in limbs and, consequently, is not expected to uncover TREEs driving gene expression in other limb tissues, including the wound epidermis. A comprehensive, high-quality epigenetic profiling of axolotl limb regeneration would provide the ideal dataset for comparative studies.

In conclusion, our study establishes *Polypterus* as a robust model that complements zebrafish by offering unique insights into the mechanisms underlying complete fin regeneration. Our findings highlight the power of comparative studies incorporating non-traditional research models to distinguish between ancestral and derived regenerative processes. Ultimately, a broad comparative approach that includes limb regeneration-incompetent species—such as post-metamorphic frogs, lizards, and mice—will be instrumental in

identifying physiological, cellular, genetic, or epigenetic features that may have been modified or lost in amniotes.

## Methods

### Animal work

*Polypterus* (*P. senagalus*) and axolotls (*Ambystoma mexicanum*) specimens were maintained and used in accordance with an approved Louisiana State University (LSU) IACUC protocol IACUCAM-25-047. Juvenile fish were obtained from commercial vendors and maintained in individual tanks in a recirculating freshwater system at 27–28 °C and a day-night cycle as 12 h light/12 h dark. Juvenile axolotls were obtained from the Ambystoma Genetic Stock Center at the University of Kentucky and maintained in individual tanks with 40% Holtfreter's solution at 18–19 °C, and a day-night cycle as 12 h light/12 h dark. Fish were anesthetized in 0.05% MS-222 (Sigma), diluted in fish system water, before amputations. Pectoral fins of *Polypterus* fish ranging from 7 to 10 cm were bilaterally amputated across the proximal endoskeleton using a sterile scalpel blade. Regenerating fins were harvested at 1, 3, and 7 dpa and processed according to downstream protocols. For uninjured fin samples, a portion of the amputated fins encompassing the endoskeletal elements was sampled and processed. As we anticipate future comparisons of some of the sequence datasets described here with datasets generated from animals treated with specific protein inhibitors, regenerating fins were harvested from DMSO-treated animals (0.1% in fish system water) for bulk RNA-seq, spatial transcriptomics, and snRNA-seq experiments. For axolotl samples, animals ranging from 8 to 12 cm were anesthetized in 0.05% MS-222, diluted in 40% Holtfreter's solution, and had their forelimbs amputated at the level of the upper arm. Regenerating limbs at 3, 7, and 14 dpa were harvested and processed for spatial transcriptomics. For uninjured limb samples, a portion of the amputated limb encompassing the upper arm was sampled and processed. All animals were euthanized according to our approved IACUC protocol (0.25% MS-222 in the appropriate animal media).

### Histology

Uninjured and regenerating tissues were flash-frozen using isopentane (2-methylbutane) chilled with liquid nitrogen. Isopentane was poured into a metal canister, which was placed in liquid nitrogen to equilibrate. Fresh tissues were embedded in Tissue-Tek O.C.T. compound (Sakura) in appropriate molds. Using forceps, the molds containing O.C.T. compound were placed in the equilibrated isopentane until completely frozen. Samples were stored at −80 °C and later sectioned (10 µm thickness) in a Leica CM1520 cryostat. Slides were stored at −80 °C until subsequent use. Hematoxylin and Eosin staining was performed as previously described[3]. H&E slides were visualized and imaged using a Nikon Eclipse N*i* microscope with a Nikon DS-Fi3 camera.

### HCR-FISH.

HCR-FISH was performed on 10 µm fresh frozen tissue sections following the manufacturer's protocol (Molecular Instruments). Tissue sections were stored at −80 °C until use. Probes for this study were generated by Molecular Instruments and can be ordered using the following lot numbers: RTT023 (for *col6a3*) and SVB695 (for *krt17*).To begin, the sections were fixed in ice-cold 4% paraformaldehyde (PFA) (MP Biomedicals) in phosphate-buffered saline (PBS) for 15 min at room temperature. After fixation, the slides for *col6a3* probes were washed three times with 1X PBS for 5 min each. For *krt17* probe, slides were submitted to a dehydration series (5 min immersion in 50%, 70%, and 2 times in 100% EtOH), followed by a PBS wash. Prehybridization was then performed at 37 °C for 10 min. Probe hybridization was carried out overnight at 37 °C. Following the hybridization, the slides were washed three times with probe wash buffer at 37 °C for 15 min each, and then two washes were performed with 5X SSC-T buffer at room temperature for 5 min each. Pre-amplification was done

by applying amplification buffer to the slides for 30 min at room temperature. Hairpins (H1 and H2) were denatured at 95 °C for 90 s and allowed to cool to room temperature for 30 min in the dark. The hairpins were diluted 1:50 in amplification buffer and applied to each slide. The slides were incubated overnight at room temperature. After incubation, the slides were washed twice with 5X SSC-T buffer for 15 min each (*col6a3*) or 4 × 15 min washes in HCR Gold amplifier buffer (*krt17*). ProLong Gold with DAPI (Invitrogen, #P36925) was used as mounting media and for nuclear counterstaining. Finally, the slides were stored at 4 °C in the dark until imaging.

### Bulk RNA-sequencing and analysis

For RNA-sequencing, total RNA was extracted from uninjured and 3 dpa pectoral fin tissue (3 biological replicates, 1 pair of fins per replicate) using TRIzol reagent (Invitrogen) according to the manufacturer's protocol, followed by DNase I treatment and column clean-up (Qiagen RNeasy kit) to remove any residual DNA. mRNA-enriched libraries were generated and sequenced at Novogene (https://www.novogene.com/us-en). Briefly, mRNA purified using poly-T oligo-attached magnetic beads was used to generate sequencing libraries using the NEBNext Ultra RNA Library Prep Kit for Illumina, following the manufacturer's recommendation. Sequencing was performed on an Illumina Novaseq 6000 platform to produce 150 bp paired-end reads. Sequence reads were mapped to *P. senegalus* Genome Assembly ASM1683550V1 using STAR V2.7.10b[95]. Read count extraction was performed using featureCounts v2.0.6[96]. Read count normalization and differential gene expression analysis were performed using DEseq2 R package[97]. Genes with a normalized count value < 10 per row were filtered out. Differentially expressed genes were defined by a log2Fold-Change > 1.0 or <−1.0 with a False Discovery Rate (FDR) < 0.05. For subsequent comparative sequence analyses, a subset of *Polypterus* and axolotl genes was manually annotated according to sequence similarity and/or synteny to other vertebrate genomes (Supplementary Data 10).

### Spatial transcriptomics

Cryosections (10 µm thickness) from regenerating *Polypterus* pectoral fins (uninjured, 1, 3, and 7 dpa) and regenerating axolotl forelimbs (uninjured, 3, 7, and 14 dpa) were placed onto 10X Genomics Visium Spatial Gene Expression Slides. Frozen tissue blocks were prepared as described in the Histology section. Visium slide samples were processed using the 10X Genomics Spatial Gene Expression kit (#1000188) according to the instructions in the Visium Spatial Gene Expression User Guide. Tissue permeabilization was performed for 18 min as previously determined using a 10X Genomics Visium Tissue Optimization Slide (#1000193). Brightfield histology images were obtained on a Leica DM6B Microscope at 10X magnification, using a Leica DFC450 color CCD camera, and stitched using the Leica Application suite X software (LAS X). Generated libraries were sequenced at Novogene on an Illumina NovaSeq 6000 platform (paired-end, 150 bp), with the sequencing depth of 50,000 reads per tissue covered spot. Sequence reads were mapped to *P. senegalus* Genome Assembly ASM1683550V1 or *Ambystoma mexicanum* Genome Assembly UKY_AmexF1_1. Raw sequence data (FASTQ) and histology images were processed using Space Ranger 3.0.1. Data visualization and generation of specific gene expression images were performed using Loupe Browser 8.0.0. Basic Seurat functions were used for generation of elbow plots in RStudio Version 2023.06.1 + 524.

### Nuclei preparation and single-nucleus RNA-sequencing

Uninjured and regenerating pectoral fins at 1, 3, and 7 dpa were dissected using sterile scalpel blades and spring scissors. For each time point, a pair of pectoral fins from two individuals was collected and pooled as one sample after nuclei isolation. A total of two biological replicates (each comprising two polled animal samples) were prepared

for each time point. Pectoral fin tissue from each animal was minced first with a spring scissor and then with a razor blade in a petri dish on ice. The minced tissue was resuspended in 1 mL of 0.3X lysis buffer (lysis buffer provided in the Active Motif ATAC-seq kit, #105320, diluted 1:3 with a lysis dilution buffer: 10 mM Tris-HCl, pH: 7.5, 50 mM NaCl, 20 mM Mg2Cl, 10 % BSA) and then homogenized in a pre-chilled Dounce homogenizer (12 strokes). The homogenate was filtered using a 40 μm cell strainer and immediately diluted with additional 4 mL of lysis dilution buffer and then centrifuged at $400 \times g$ at 4 °C for 5 min in a swinging bucket rotor. The pellet was quickly washed with 1 mL of wash buffer (1X PBS/0.5% BSA), containing 40 U/μl of Protector RNAse inhibitor (Millipore-Sigma #3335399001), without resuspension, and then resuspended in 0.5 mL of wash buffer with RNAse inhibitor. At this point, samples from the two individuals were combined. Nuclei were counted using a hemocytometer, and then a volume corresponding to $10^6$ nuclei was transferred to an Eppendorf LoBind microcentrifuge tube and centrifuged at $400 \times g$ at 4 °C for 5 min. Nuclei were fixed and frozen using the Evercode Nuclei Fixation Kit v3 (#NF100, #NF200) from Parse Biosciences following the manufacturer's protocol. Fixed and frozen nuclei were then processed to generate barcoded libraries aiming for 7000 nuclei per sample using the Evercode WT v3 Kit (#WT100A-D, #WT200) as per manufacturer's instructions. Resulting libraries were sequenced by Novogene at the depth of 20,000 reads per nuclei in an Illumina Novaseq Xplus platform (paired-end, 150 bp).

## Single-nucleus RNA-seq analysis

Sequence reads were mapped to *P. senegalus* Genome Assembly ASM1683550V1. SnRNA-seq FASTQ files were processed using the Trailmaker™ pipeline module (https://app.trailmaker.parsebiosciences.com/; v1.4.0, Parse Biosciences, 2024). In this module, sequences were processed for demultiplexing, barcode correction, read alignment, and transcript quantification. Quantified transcripts were used to generate a cell-by-gene count matrix that was automatically redirected to the Insight module of Trailmaker for downstream analysis. During the Insight module processing, background, doublets, and poor-quality nuclei (high content of mitochondrial gene detection and low level of detected transcripts) were excluded using the Trailmaker default data automatic processing. For our dataset, the minimum number of transcripts per cell varied from 390 to 928, and the median number of transcripts per cell after all filters ranged from 1909 to 5199 (Trailmaker sets a minimum transcript/gene number per cell/nuclei on a per-sample basis). The maximum cutoff value for maximal percentage of mitochondrial genes was 1.21%. Doublets were filtered out based on the scDblFinder method (probability threshold from 0.60 to 0.83). Data from high-quality nuclei were submitted to data normalization, principal-component analysis, and data integration using Harmony. Clusters were identified using the Leiden method, and results were visualized by Uniform Manifold Approximation and Projection (UMAP) embedding. The presto package implementation of the Wilcoxon rank-sum test was used to define cluster-specific marker genes by comparing nuclei data of each cluster to all other nuclei. Trailmaker and Seurat v5 R package[98] were used to produce heatmaps and gene expression UMAP image panels. SnRNA-seq data from zebrafish were obtained from the Gene Expression Omnibus (GEO; accession number GSE261907)[8] and reprocessed using Cell Ranger. The cellranger mkref command was used to generate the genome index, and cellranger count was employed to align the reads to the reference genome. After processing, the output files (barcodes.tsv.gz, features.tsv.gz, and matrix.mtx.gz) were uploaded to Trailmaker for downstream analyses. For the axolotl dataset[15], scRNA-seq data were downloaded from the NCBI Sequence Read Archive (SRA; project number PRJNA589484) and processed using the same pipeline. UMAPs presented through the manuscript were either created in Trailmaker or in R Studio using the Seurat FeaturePlot function. The analysis of gene set score for AER-related genes was performed using the Seurat AddModuleScore function in R Studio. The gene list used (*axin2*, *bambi*, *bmp2a*, *bmp7b*, *cd44*, *dkk1*, *dlx2a*, *fgf9*, *fgf2*, *fgf8a*, *lef1*, *msx2*, *prdm1a*, *rfng*, *rspo2*, *rspo3*, *jag2*, *sp6*, *sp8*, *fn1*, *gja1*, *lgr6*, *notch1*) was based on a previously published study[26].

## Bulk ATAC-seq library preparation

Uninjured and 3 dpa fin tissues were collected and subsequently used for nuclei isolation and tagmentation following the ATAC-seq kit protocol (Active Motif, #105320) without modifications. For each triplicate, tissue from both left and right pectoral fins was collected and pooled to comprise one biological replica. Each library was prepared using 50,000 nuclei and amplified within ten PCR cycles. After PCR amplification, the libraries were purified and sent for sequencing. The sequencing was carried out using the NovaSeq XPlus platform, generating 150 bp paired-end reads (150 million reads/sample).

## ATAC-seq data analysis

The three biological replicates of each condition were processed as follows. Raw reads were trimmed using NGmerge (version 0.3)[99]. The trimmed reads were aligned to the *P. senegalus* reference genome from NCBI (ASM1683550v1)[12] using Bowtie2 (version 2.2.5)[100]. Alignments were output as a BAM file, indexed using Samtools (version 1.18)[101]. Fragment size distribution was assessed using the ATACseqQC package (version 1.24.0)[102] in RStudio (version 4.3.1). Trimmed reads were name-sorted using Samtools (version 1.18)[101]. Genrich (version 0.6.1) was used to call peaks using the parameters -j -y -r -v -m -e -f, to remove duplicate reads, multimapped reads, and reads mapped to mitochondrial chromosome. Downstream analysis of the ATAC-seq data was performed using DiffBind (version 3.10.1) to identify differential chromatin accessibility (peaks) between uninjured and regenerating fin tissues[103]. Peak sets generated by Genrich were associated with metadata for each sample and imported into DiffBind within RStudio (version 4.3.1). Read counts were generated using summits = 75 and normalized using DESeq2 (version 1.40.2)[97]. Peaks that exhibited differential accessibility between the two conditions were selected based on a significance threshold of FDR < 0.05. Peaks were visualized using Integrative Genomics Viewer (IGV) (version 2.14.1). AnnotatePeaks.pl from HOMER (version 4.11) was used to identify and annotate the genes located nearest to peaks. Motif footprint enrichment was performed using TOBIAS[69].

## Phylogenetic analysis of HIFA homologs

The following protein sequences of HIFA homologs from representative vertebrate species were obtained from NCBI: *Ambystoma mexicanum* (HIF1A: XP_069495030.1, HIF2A: XP_069464544.1, HIF3A: XP_069500692.1, HIF4A: XP_069500864.1), *Pleurodeles waltl* (HIF1A: XP_069063495.1, HIF2A: XP_069090117.1, HIF3A: XP_069063783.1, HIF4A: XP_069063921.1), *P. senegalus* (HIF1A: XP_039598212.1, HIF2A: XP_039594325.1, HIF3A: XP_039624026.1, HIF4A: XP_039625223.1), *Protopterus annectens* (HIF1A: XP_043930427.1, HIF2A: XP_043915843.1, HIF3A: XP_043913476.1, HIF4A: XP_043931869.1), *Lepisosteus oculatus* (HIF1A: XP_069049245.1, HIF2A: XP_015218279.2, HIF3A: XP_069042482.1, HIF4A: XP_069036561.1), and *Branchiostoma lanceolatum* (HIFA: XP_066279910.1). Protein sequences were aligned using Clustal Omega v1.2.4 with default parameters, enabling MBED-like clustering for both guide tree and iteration steps. The resulting alignment in FASTA format was used as input for phylogenetic inference. Maximum-likelihood phylogenetic analysis was conducted using IQ-TREE v1.6.12 with ModelFinder to identify the best-fitting substitution model. According to the Bayesian Information Criterion (BIC), the optimal model was JTT + F + I + G4, which was subsequently used for tree reconstruction. Node support was assessed using ultrafast bootstrap with 1000 replicates. The final tree was visualized and annotated in FigTree (v1.4.4).

## Statistics and reproducibility

Three biological replicates were used for both bulk RNA-seq and bulk ATAC-seq experiments on fins at homeostasis (uninjured) and at 3 dpa. For snRNA-seq, two biological replicates—each consisting of pooled samples from two animals—were prepared for each time point. Spatial transcriptomics were performed on single tissue sections from each of the four time points analyzed for *Polypterus* fins (uninjured, 1, 3, and 7 dpa) and axolotl limbs (uninjured, 3, 7, and 14 dpa). Histology images from *Polypterus* intact and regenerating fins were selected from section series obtained from a pair of pectoral fins from two animals per stage. HCR-FISH analyses were conducted using at least two technical replicates for each of the four *Polypterus* fin time points (uninjured, 1, 3, and 7 dpa). No power analysis was used to pre-determine sample sizes. Specific details of the statistical methods and significance values used are provided in the appropriate Methods sections describing each of the RNA/DNA sequence-based analyses and in the figure legends.

## Inclusion and ethics statement

This research aligns with the inclusion and ethical guidelines embraced by Nature Communications.

## Reporting summary

Further information on research design is available in the Nature Portfolio Reporting Summary linked to this article.

## Data availability

All raw and processed data from bulk RNA-seq (GSE315989 [https://www.ncbi.nlm.nih.gov/geo/query/acc.cgi?acc=GSE315989]), spatial transcriptomics (GSE315990 [https://www.ncbi.nlm.nih.gov/geo/query/acc.cgi?acc=GSE315990] and GSE315993), snRNA-seq (GSE316120 [https://www.ncbi.nlm.nih.gov/geo/query/acc.cgi?acc=GSE316120]) and bulk ATAC-seq (GSE315999 [https://www.ncbi.nlm.nih.gov/geo/query/acc.cgi?acc=GSE315999]) have been deposited in the Gene Expression Omnibus (GEO). Processed *Polypterus* snRNA-seq data have been made available at the Broad Single Cell Portal (https://singlecell.broadinstitute.org/single_cell) (SCP: SCP3138).

## Code availability

All code used in this study is available at our GitHub repository: https://github.com/lnperez90/Sousa-et-al-2025/.

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

## Acknowledgements

We thank Patricia Schneider for helpful feedback and comments. We also thank the Advanced Microscopy and Analytical Core (AMAC) at Louisiana State University for providing equipment access and technical assistance with imaging of the spatial transcriptomics slides. This work was funded by start-up funds from Louisiana State University (I.S.), and an NSF-Integrative Organismal Systems (IOS) Enabling Discovery through GEnomics (EDGE) grant (2421117, I.S.).

## Author contributions

J.F.S., G.L., and I.S. designed research; J.F.S., G.L., H.S., and I.S. per-formed research; J.F.S., G.L., L.P., and I.S. analyzed data; J.F.S., G.L., and I.S. wrote the paper with input from all authors.

## Competing interests

The authors declare no competing interests.
