## [Peer Review file · Nature Communications]

Comparative multi-omic analysis reveals conserved and derived mechanisms of fin and limb regeneration

Corresponding Author: Professor Igor Schneider

Version 0:

Reviewer comments:

Reviewer #1

(Remarks to the Author)

Sousa et al., investigated in this work whether there are shared mechanisms of regeneration between *Polypterus senegalus* fin regeneration and axolotl limb regeneration. In this study, the authors describe the following observations: By using single-nucleus RNA sequencing on an early time of regeneration, the authors delineate the cell populations of *Polypterus senegalus* fin regeneration. Complemented with spatial transcriptomics of one time point in both *Polypterus senegalus* and axolotl, the authors compared expression of genes in wound epithelium and blastema. The authors also observe processes common to regeneration such as a pro to anti-inflammatory switch, DNA repair and hypoxia response. Overall, we are supportive of this study, since this could help to uncover the evolutionary origin of limb and fin regeneration. There are several advantages of using *Polypterus senegalus*: their divergence from teleosts before the latter's WGD, their structural components in the fin that share characteristics with the limb bones of tetrapods, and beyond the usefulness of species comparison, the use of *Polypterus senegalus* can inform on ancestral vs derived mechanisms of regeneration. We believe however, that at present, many of the findings are too purely descriptive in nature, and this work could thus benefit from some additional evidence and analysis to support the claims. Particularly, how knowledge of the regenerative process in additional species improves our understanding of what is important for regeneration in vertebrates to occur, and how it has evolved over time. The details are listed below.

1) One interesting piece of information is the description of the cell populations present at the early time points. The authors noted a significant presence of erythrocytes and investigated some growth factors and ECM modulators and conclude that they do not contain the secretory profile that has been described before. However, from the cellular landscape, important information about these dynamic changes could be drawn. For example, 1) what are the erythrocytes expressing besides the few factors they looked at? Later in the manuscript, the expression of a HIF1a-like gene in erythrocytes seems to be unique of *Polypterus senegalus*. Characterizing this population could inform more on their function or relevance in regeneration. 2) what cell types are proliferating?

2) The authors mention that in Japanese fire-bellied newts, erythrocyte clumps express genes of secreted growth factors and matrix metalloproteases that might have a role in limb regeneration. Meanwhile, they find that this is not the case in *Polypterus* erythrocytes. However, as they have data of 3 dpa axolotl, it would be interesting for the authors to show if the axolotl erythrocytes are more like the newt or like the bichir. Additionally, in the mentioned newt paper (<https://doi.org/10.1038/s41598-018-25867-x>), the growth factor secreting erythrocytes largely appear around 14 dpa (stage II blastema) and 27 dpa (stage III blastema). Thus, could it be that the *Polypterus* samples in the paper are simply too early to detect these erythrocyte clusters?

3) In Figure 2, the authors mention that col17a1b was previously identified as a marker for wound epidermis, however the staining shows that it is also expressed in intact epidermis and not necessarily specific of a wound epidermis. Also, col6a3 is proposed here as a marker for blastema cells and they show HCR detection in mesenchyme, but there is not a good overall expression view. Since they claim the expression of col6a3 indicates that connective tissue cells are major contributors to the blastema. We would suggest better images of these markers or complementing with their spatial transcriptomics data.

4) The time (3dpa) to compare both species, is chosen based on the presence of blastema cells and multilayered wound

epidermis. This is not convincing for the following reasons: axolotls of this size, have almost no cells accumulated below the wound epidermis at this timepoint. The wound epithelium is still few cells in thickness compared to the histological section shown for *Polypterus senegalus* and the accumulation of cells below it. What is here marked as thickened wound epithelium in the axolotl appears to be the mature epidermis. The protruding bone and the normally retracting connective tissue create this shape that can be confused with a blastema. This could explain for example the mismatch in gene expression in some of the genes of interest, because the axolotl sample corresponds to an earlier time point in *Polypterus senegalus*. We would suggest a validation of these stages. By using either their own data and time datasets in doi: 10.1126/science.aaq0681 and doi: 10.1371/journal.pcbi.100 , or those of others, the authors could provide a conclusive stage comparison.

5) The authors expand on the species comparison of wound epithelium and blastema, however in Figure 4, they reduce their comparison to a few genes. Given their data, a more comprehensive analysis could provide valuable information. For example, they could provide the list of their top 50 DEG in the blastemas and wound epidermis for both species and build from there a hypothesis on the species-specific, or conserved mechanisms of regeneration. Interestingly, there are 2 blastema clusters in *Polypterus senegalus*, but no mention in the text about them. There seem to be a proximal and a distal field, which could be an exciting result.

6) A strength discussed in this paper of using the *Polypterus senegalus* as a model of appendage regeneration, and one which we strongly agree with, is that it did not undergo the whole-genome duplication that zebrafish and other teleosts did (tsWGD), and thus many of its genes have 1:1 orthology with tetrapod genes. With this stated benefit, and with how many *Polypterus senegalus* genes are talked about, we wonder if the authors are missing an opportunity to formalize the gene naming conventions in non-teleost actinopterygians. For example, there are cases where gene names have already been assigned a name in the current RefSeq annotation of *Polypterus senegalus*, such as *pax7a* and *col17a1b*, and the authors keep these names. However, the “a” and “b” copy naming convention comes from the zebrafish, in which the ancestral gene was duplicated in the tsWGD, and thus the *Polypterus senegalus* is not expected to possess a *pax7b* or *col17a1a*. Meanwhile, the authors newly annotate at least 18 genes that were previously only given LOC IDs, and for these genes they assign tetrapod-like gene names, i.e., no “a” or “b” versions. However, in the case of the manually annotated genes *esam*, *tgfb1*, *fn1*, *fgf10*, *lima1*, and *tnc*, zebrafish have “a” and “b” versions of each of them. Therefore, if the goal is to keep consistency with the naming convention of already annotated genes, one of these versions would be selected and used as the gene ID, e.g., *esama* or *tgfb1b*. While it is difficult to challenge already assigned gene IDs, the discussion of so many different genes in the paper, with many being annotated for the first time, could make this a good opportunity to establish names in the *Polypterus senegalus* that make orthology with the axolotl and other tetrapods clearer, and thus further highlight the benefit of having fish models that are not teleosts.

Additionally, the axolotl gene nomenclature is not following the generally agreed upon style of the salamander community (<https://doi.org/10.1002/dvdy.351>), and so should be changed. The current style in the paper also makes it sometimes unclear if what is being discussed is a protein or a gene due to not using italics for axolotl genes.

7) Is there a reason that the authors do not investigate the spatial location of pro-inflammatory and anti-inflammatory genes in their spatial transcriptomic data? They show that in the snRNA-Seq data, these genes are expressed by different populations of cells, and thus it would be interesting, and potentially informative, to see if these different populations are also located in different areas of the regenerating fin.

8) In Figure 9, it is not really clear why the genes discussed here stood out or were selected for further examination. Could be better if the authors stated something like that *tfpi2* was selected for study because they were interested in seeing what genes were specific to *Polypterus* regeneration. Conversely *tnfaip6* seems to have been investigated because it's expressed in the mesenchymal cells of amniotes and teleosts, demonstrating a strong conservation of its role over vertebrate evolution. By more explicitly stating the reasons that drove the selection of these genes, it could enable the authors to more clearly highlight the benefits of studying regeneration across species, and to later discuss the ways in which future experiments are impacted by the knowledge of if gene roles in a particular species are derived or conserved.

Minor comments:

1) In Figure 1c, a color-coded correspondence of structures between species could be very helpful for people not familiar with the species. For example: in the endoskeleton of the *Polypterus senegalus* fin what corresponds to humerus, radius, carpals, phalanges, etc. in the axolotl.

2) There is a reference to figure (Supplementary Fig. 4c and d), however, there is no “d” panel

3) Page 9: “Contrary to mammals, urodele amphibians such as newts can regrow severed limbs even at high environmental oxygen concentrations.”

This sentence makes it sound like mammals can regrow severed limbs.

4) Page 8: “Unlike *dif1bb*, *lima1* and its axolotl ortholog *LIMA1* were expressed... (Fig. 10d).”

dif1bb is the wrong gene to say here. The figure states “*DIP2B*”, and *dif1bb* is not mentioned anywhere else in the paper.

5) Page 6: “A previous study showed that among *Polypterus* myoglobin-like proteins, *PseMb1* the product of LOC120537046, termed *PseMb1*,”

The phrasing of how *PseMb1* was given its name should be improved.

6) It is not clear if the authors imply that previous studies which found that Frem2 is a marker of wound epidermis cells in the axolotl limb have assigned the gene ID incorrectly, and that it is actually Frem3. It would be good if the authors are clearer about their thoughts on axolotl Frem2/Frem3, and how its expression compares to Polypterus.

7) A comment on comparing single-nucleus transcription (de novo transcription) to spatial transcriptomics (whole cell transcripts), would be appreciated.

8) The authors use the word integration of data across the manuscript to refer to how they use sn-sequencing data and spatial transcriptomics. This could be misleading as the data was not integrated but rather is analysed side-by-side.

Reviewer #3

(Remarks to the Author)

This was an interesting paper that establishes the fish *Polypterus senegalus* as a new model for studying regeneration in the fin. Although zebrafish have been used for this purpose, only the dermal rays of the zebrafish fin can regenerate, which are different from the endochondral bones present in limbs. *Polypterus* fins, on the other hand, have more extensive endochondral bones that are capable of regenerating, making them a better model for comparative studies of fin and limb regeneration. Here, the authors use a variety of multi-omics approaches, including single-nucleus RNA sequencing, spatial transcriptomics, and ATAC-sequencing to identify commonalities and differences between regeneration in axolotl limbs and *Polypterus* fins.

Overall, the contribution of a new organism for studying the process of regeneration is significant. The authors present several impressive datasets and make great strides in establishing *Polypterus* as a useful model. Although many of the genes and pathways identified have been shown in other systems, the authors do a good job of explaining the benefits of studying regeneration in *Polypterus* and identify some unique genes and pathways that warrant further study in the future. There are no concerns about the methodology, and this is an impressive amount of data. However, the context for the extensive results was sometimes lacking. This made it difficult to understand the conclusions the authors were drawing and to assess the validity of these conclusions. Please see some minor clarifications to address below.

Minor clarifications:

1. Please label the figures (Fig.1, Fig. 2, etc). It was difficult to toggle back and forth between the manuscript, figure legends, and figures when the figures had no labels. It would also be helpful to have the lines numbered in the manuscript.
2. An issue that was recurrent throughout the manuscript was a lack of context for the findings. Many sections in the results section could use at least one clarifying sentence. Please see some select examples below.
3. In the first results section (“the cellular landscape of *Polypterus* fin regeneration”) the authors mention that erythrocytes are not important in *Polypterus* for fin regeneration while they are in salamanders. Are erythrocytes also not important in zebrafish fin regeneration and therefore the role of erythrocytes is limb specific? In the discussion section, the authors state that this warrants further study, but more discussion about this would be useful.
4. For the next section (“cellular contributions to the *Polypterus* fin wound epidermis and blastema”) there is a very nice and thorough description of the role of connective tissue cells in the *Polypterus* fin blastema, but no context provided. Is this similar in other fish and limbs? Or different?
5. I thought that the sections about hypoxia and the DNA damage response were the clearest to understand. The authors did a good job of describing the role of both pathways in other organisms and clearly identify *Polypterus* specific pathways and genes that warrant more study in the future.
6. In the section “epigenetic analysis uncovers candidate fin regeneration-responsive elements” it was unclear if the authors are hypothesizing that Peak 5 is driving *lima1* expression instead of *dip2bb*. Please state this clearly if that is the case. For example, for the sentence “While both genes are upregulated over 2-fold, bulk RNA-seq showed that *lima1* expression at 3 dpa is over 15 times greater than that of *dip2bb*” please add a sentence such as “indicating that Peak 5 is likely driving expression of *lima1*.” If that conclusion is not correct, please state what the correct conclusion is from this section about *dip2bb* and *lima1*.
7. It would be very helpful to have some sort of table identifying which genes and/or pathways are conserved and diverged between *Polypterus*, salamanders, and other fish. Do the authors think that regeneration is more conserved or diverged between *Polypterus* and salamanders? What about between *Polypterus* and zebrafish? Many genes and pathways were discussed, and it was difficult to get a sense of how conserved this process is between the different organisms.

Reviewer #4

(Remarks to the Author)

In the work from Sousa et al., the authors leverage evolutionary relationships across the vertebrate tree of life to compare limb regeneration by the Urodele axolotl salamander with that of a more closely related polypterid, *Polypterus senegalus*. The endoskeletal fin regeneration of *Polypterus* is a closer comparison than teleost regeneration of dermal bony fins. This manuscript builds on the Schneider lab’s prior work establishing *Polypterus*’ regenerative ability and evolutionary relationship relative to axolotl and other fish lineages (Darnet, et al., 2019, PNAS). In this manuscript, the authors perform a full multi-omic comparison of axolotl limb and *Polypterus* fin regeneration that encompasses single nuclei RNA-seq (snRNA-seq), Visium spatial transcriptomics at 3 days post amputation (3dpa), and bulk Assay for Transposase-Accessible Chromatin sequencing (ATAC-seq). The authors identify a number of similarities and differences in the transcriptional and genomic regulation immediately following amputation. The rigor associated with having both limb and fin tissues subjected to the same experimental methodologies and analyses is a great resource for the regeneration field and represents an

excellent standard for future studies.

Despite the technical standard that this work represents, I struggled to find cohesive conclusions from what seemed like observational “vignettes” picked from the multi-omic datasets. I also was unclear if, or how, this work related to their prior analysis of limb and fin regeneration (Darnet et al.). I don’t think this should detract from the work, but I do think there are a few meaningful changes that could create a more focused and presentable manuscript.

- My primary critique is that this work should be presented primarily as a technical resource since none of the individual results sub-sections are comprehensive or mechanistic enough to make major conclusions. In that vein, I would suggest that the authors put effort into making the data accessible to readers via an explorable R-Shiny site or uploading data to the Broad Institute Single Cell Browser (or equivalent). This would translate the technical prowess in the manuscript to a highly impactful resource for the regeneration community.

- One of the primary arguments put forth by the authors for using the Polypterus fin as a comparative model to the axolotl limb, in contrast to the more ubiquitous zebrafish fin, is the presence of endoskeletal regeneration. Yet, the authors make no connections between regeneration of the fin skeleton and axolotl limb bones. This seems like a large oversight, and it is not clear if it is because of technical limitations such as not being able to detect skeletal cell types in snRNA-seq data (no clusters were described). I recognize that the timepoints the authors examined are early after amputation and do not represent timepoints that skeletal patterning is happening, however a comparison of skeletal cell types and signals that influence processes like bone histolysis, and skeletal progenitor signaling could be examined even at early phases.

- Related to the point above, the Discussion section of the manuscript is used primarily as a re-hash of result subsections rather than an opportunity in the manuscript to describe limitations, future outlook/directions, and postulating on a more cohesive overview of the work (e.g., in what ways is Polypterus regeneration more similar to teleost regeneration and what ways is it more like limb regeneration?). I appreciate that the authors try to put some of their results in context of prior literature, but it is sometimes hard to follow with non-precise reference to other papers. For example, the connection of Kazald expression to other systems was not very clearly articulated and I’m not sure if the reference given really provides the evidence that the authors indicate in the discussion text.

- As a reader, one of the most interesting and relatively untouched aspects of this work is the difference in spatial distribution of gene expression in the proximal-distal axis between the two organisms. The authors point this out relative to DNA damage markers between Polypterus and axolotl, but this seems evident in many other gene comparisons between fin and limb in the manuscript’s figures. Taking an unbiased gene set (e.g., top 100 DEGs) and determining the relative proximal-distal expression zone (using some standard of normalization and distance) would be a very interesting and quantitative comparison using spatial transcriptomic data. This would really leverage the author’s rigorous data in a way that is novel and unlikely to be represented in other manuscripts.

- Finally, I have a hard time really agreeing on the author’s definitions of “wound epidermis” markers derived from snRNA-seq in Figures 1-3. The authors describe “wound epidermis” clusters, but these clusters seem to exist in the uninjured data set as well, just with fewer cells represented relative to post amputation. They describe col17a1b as a marker of these epidermal cells, but again, there are col17a1b-expressing cells before regeneration as well. It is not until Figure 5 when markers are connected between Visium and the snRNA-seq clusters do you observe marker genes that are expressed specifically after amputation in wound epidermis snRNA-seq clusters. Markers like col17a1b should be clarified as upregulated or “amplified” during regeneration instead of markers that uniquely identify the wound epidermis. In general, I think this double standard for wound epidermis markers is confusing and I would simply suggest moving the col17a1b data to supplemental data since the data in Figure 5 is much more convincing although it lacks high resolution HCR micrographs.

Version 1:

Reviewer comments:

Reviewer #1

(Remarks to the Author)

The corrections submitted by Sousa et al., have greatly enhanced this work. The authors have thoughtfully addressed the comments and it was a pleasure to help refine it. No further comments to make.

Reviewer #5

(Remarks to the Author)

Point by point response to reviewers

We sincerely thank the reviewers and editors for their thoughtful and constructive feedback on our manuscript, now entitled “Comparative multi-omic analysis reveals conserved and derived mechanisms of fin and limb regeneration.” We greatly appreciate the careful evaluation and insightful suggestions, which have substantially improved the clarity, depth, and rigor of our study. In this revised version, we have generated additional spatial transcriptomic datasets, expanded comparative analyses across *Polypterus*, axolotl, and zebrafish, refined cell-type annotations, and clarified key aspects of our results and interpretations. We have addressed all comments in detail below, with point-by-point responses and corresponding changes indicated in the revised manuscript.

Reviewer Comments

Reviewer #1 (Remarks to the Author):

1) One interesting piece of information is the description of the cell populations present at the early time points. The authors noted a significant presence of erythrocytes and investigated some growth factors and ECM modulators and conclude that they do not contain the secretory profile that has been described before. However, from the cellular landscape, important information about these dynamic changes could be drawn. For example, 1) what are the erythrocytes expressing besides the few factors they looked at? Later in the manuscript, the expression of a HIF1a-like gene in erythrocytes seems to be unique of *Polypterus senegalus*. Characterizing this population could inform more on their function or relevance in regeneration. 2) what cell types are proliferating?

We thank the review for prompting us to looking further into these erythrocytes. We now have analyzed the recruitment of these cells in *Polypterus*, axolotl and zebrafish and added a results section dedicated to this matter, entitled: “Limb and fin regeneration followed by endoskeleton-level amputation is marked by robust expansion of *hif4a*⁺ erythrocytes”. (starting at line 369). We show that the emergence of these cells is not a hallmark of zebrafish tail fin ray regeneration. Our analysis of the top 100 differentially expressed genes in erythrocytes of *Polypterus* and axolotl confirmed the overall lack of ECM remodeler or secreted morphogen expression and showed typical as well as atypical erythrocyte genes commonly and uniquely expressed in these species. A careful reassessment of these datasets revealed the erythrocyte-specific enrichment of a HIF4A paralog and phylogenetic analysis supports that these genes encode HIF4A ortholog. The presence of *Hif4a*-expressing erythrocytes during regeneration in *Polypterus* and axolotl—but not zebrafish tail fin ray—possibly consists in a unique adaptation to the metabolic and vascular demands of regenerating an endoskeleton-bearing appendage. These analyses are shown in a new figure (Fig. 8), new supplementary data (Supplementary Fig. 8) and table (Supplementary Data 8).

In addition, we have added a paragraph to results section entitled “The cellular landscape of *Polypterus* fin regeneration”, where we clarify that the proliferating cell 1 and 2 clusters present epithelial cell features (expression of *tp63* and *evpl*), and expression of *col6a3* in proliferating cell 3 cluster suggests that this cluster constitutes connective tissue proliferating cells (line 141).

2) The authors mention that in Japanese fire-bellied newts, erythrocyte clumps express genes of secreted growth factors and matrix metalloproteases that might have a role in limb regeneration. Meanwhile, they find that this is not the case in *Polypterus* erythrocytes. However, as they have data of 3 dpa axolotl, it would be interesting for the authors to show if the axolotl erythrocytes are more like the newt or like the bichir. Additionally, in the mentioned newt paper (<https://doi.org/10.1038/s41598-018-25867-x>), the growth factor secreting erythrocytes largely appear around 14 dpa (stage II blastema) and 27 dpa (stage III blastema). Thus, could it be that the *Polypterus* samples in the paper are simply too early to detect these erythrocyte clusters?

This was a very pertinent point raised by the reviewer. Given additional comments made by this and other reviewers we opted to expand our single cell and spatial transcriptomics analyses by 1) generating additional spatial RNA-seq datasets for *Polypterus* (now uninjured, 1 dpa, 3 dpa and 7 dpa) and axolotl (now uninjured, 3 dpa, 7 dpa, and 14 dpa); and 2) by analyzing existing axolotl single cell dataset (stages

uninjured, 3 dpa, 7 dpa, 14 dpa, and 22 dpa). In contrast to the post-metamorphic newt, where *Newtic1*⁺ erythrocyte clumps seem to express secreted factors and matrix metalloproteases during limb regeneration, our analyses indicate a distinct erythrocyte profile in *Polypterus* and axolotl. We found no enrichment of secreted growth factors or extracellular matrix remodelers in erythrocytes from either species (even when later stages are considered), suggesting that these cells are unlikely to contribute directly to pro-regenerative signaling. Moreover, while *Newtic1* expression in adult newts is associated with erythrocyte aggregates, the axolotl *Newtic1* ortholog is expressed almost exclusively in epidermal cells in both scRNA-seq and spatial transcriptomic datasets, with no detectable expression in erythrocytes, which resembles what has been described for larval newts. Together, these findings support the conclusion that *Newtic1*⁺ erythrocyte clumps are a post-metamorphic salamander feature, and that erythrocytes in *Polypterus* and axolotl instead exhibit strong *Hif4a* expression, consistent with a role in hypoxia adaptation rather than secretion of regenerative cues. These results are now described in the section entitled: "Limb and fin regeneration followed by endoskeleton-level amputation is marked by robust expansion of *hif4a*⁺ erythrocytes". (starting at line 369).

3) In Figure 2, the authors mention that *col17a1b* was previously identified as a marker for wound epidermis, however the staining shows that it is also expressed in intact epidermis and not necessarily specific of a wound epidermis. Also, *col6a3* is proposed here as a marker for blastema cells and they show HCR detection in mesenchyme, but there is not a good overall expression view. Since they claim the expression of *col6a3* indicates that connective tissue cells are major contributors to the blastema. We would suggest better images of these markers or complementing with their spatial transcriptomics data.

We thank this and other reviewers for drawing our attention to the matter of *col17a1* expression in our study. This has prompted us to generate the additional spatial transcriptomics datasets for both species, as well as incorporating analyses of publicly available axolotl limb regeneration and zebrafish tail fin regeneration single cell/nucleus RNA-seq datasets, which greatly enriched our study. A more in-depth analysis of wound epidermis cells and candidate markers is now reported in results section entitled "Reactivation of apical ectodermal ridge-like gene programs in the *Polypterus* wound epidermis". (starting at line 225). Whereas *col17a1* does become upregulated in the distal wound epidermis, it is expressed in homeostasis and throughout regeneration in all species and therefore is now more accurately described in our manuscript as a basal epidermis marker. A refined analysis of genes differentially upregulated in the presumptive wound epidermis is now provided and shows a more established marker of activated keratinocytes (*krt17*) being conspicuously upregulated in the distal wound epidermis in both single nucleus RNA-seq and spatial transcriptomics datasets. HCR-FISH for *krt17* was now added and largely mirrors the pattern we observe in our *Polypterus* spatial transcriptomics data. Additional markers including *cyp27b1* and *mmp13* are also shown, as are comparisons of gene expression patterns with axolotl and zebrafish. These data resulted in new figures and tables, specifically: Fig.4 and 5, Supplementary Figs.5, 6 and 7, and Supplementary Data 2.

With regards to *col6a3*, we have made the following changes to accommodate this reviewer's appropriate concerns: 1) we have now clarified in the results (section entitled "Blastema cell heterogeneity", starting at line 299) that *col6a3* is being reported not as a blastema marker but as a general connective tissue (CT) cell marker in all species, which is expressed in all stages analyzed but becomes enriched distally in the blastema region indicating migration of some CT cells to this area; 2) We have added improved HCR panels to show *col6a3* expression at homeostasis and during *Polypterus* fin regeneration; and 3) the addition of spatial transcriptomics datasets at later stages improved the visualization of the expression dynamics of *col6a3* as well as other markers of subsets of CT cells.

4) The time (3dpa) to compare both species, is chosen based on the presence of blastema cells and multilayered wound epidermis. This is not convincing for the following reasons: axolotls of this size, have almost no cells accumulated below the wound epidermis at this timepoint. The wound epithelium is still few cells in thickness compared to the histological section shown for *Polypterus senegalus* and the accumulation of cells below it. What is here marked as thickened wound epithelium in the axolotl appears to be the mature epidermis. The protruding bone and the normally retracting connective tissue create this shape that can be confused with a blastema. This could explain for example the mismatch in gene expression in some of the genes of interest, because the axolotl sample corresponds to an earlier time point in *Polypterus senegalus*. We would suggest a validation of these stages. By using either their own

data and time datasets in doi: 10.1126/science.aag0681 and doi: 10.1371/journal.pcbi.100 , or those of others, the authors could provide a conclusive stage comparison.

This is a valid concern that we now address this by generating new spatial transcriptomics datasets for later stages of axolotl limb regeneration (stages added: uninjured, 7 dpa and 14 dpa), as well as by incorporating publicly available scRNA-seq data of axolotl limb regeneration. At 7 dpa and 14 dpa the thickened wound epidermis is morphologically visible and shows the expression of established WE/apical epithelial cap markers. While interspecies stage-to-stage matching is difficult to achieve, our datasets are now extended to provide the general dynamics spanning homeostasis, wound closure, AEC formation and blastema formation in both species.

5) The authors expand on the species comparison of wound epithelium and blastema, however in Figure 4, they reduce their comparison to a few genes. Given their data, a more comprehensive analysis could provide valuable information. For example, they could provide the list of their top 50 DEG in the blastemas and wound epidermis for both species and build from there a hypothesis on the species-specific, or conserved mechanisms of regeneration. Interestingly, there are 2 blastema clusters in *Polypterus senegalus*, but no mention in the text about them. There seem to be a proximal and a distal field, which could be an exciting result.

We agree with the reviewer and have expanded our analyses on wound epidermis and the connective tissue cells contributing to the blastema, leveraging the newly generated spatial transcriptomics datasets and publicly available single cell data for axolotl and zebrafish. Differential gene expression analysis enabled us to 1) identify high-confidence markers for the wound epidermis; 2) identify shared and derived patterns of wound epidermis gene expression in *Polypterus* and axolotl; and 3) identify a comprehensive list of genes enriched in the AEC/blastema of *Polypterus*, axolotl and zebrafish. These data have replaced the previous Fig. 4, and are featured in section entitled “Reactivation of apical ectodermal ridge-like gene programs in the *Polypterus* wound epidermis” (starting at line 225), in the new Figs. 4, 5 and 6, Supplementary Figs. 5, 6 and 7, and Supplementary Data 2.

Furthermore, we refined our cluster analysis on spatial transcriptomics data and now provide novel findings regarding the two blastema clusters, which we refer to as proximal and distal CT cell populations. We had initially worked with default parameters for clustering, which led to overclustering of spatial data. We corrected this now by applying within-cluster sum of squares (WCSS) as a metric of clustering compactness. This resulted in fewer clusters that are more likely to be biologically meaningful. Even so, both *Polypterus* 7 dpa and axolotl 14 dpa showed proximal and distal CT compartmentalization with k-means clusters = 5. Analysis of the top 100 genes expressed in these territories revealed clear distinguishing markers and other features of these subdivided cell populations conserved between *Polypterus* and axolotl. These new analyses are described in a new section entitled “Emergence of proximal-distal organization within regenerating connective tissue” (starting at line 336), and featured in newly produced Fig. 7, and Supplementary Data 3.

6) A strength discussed in this paper of using the *Polypterus senegalus* as a model of appendage regeneration, and one which we strongly agree with, is that it did not undergo the whole-genome duplication that zebrafish and other teleosts did (tsWGD), and thus many of its genes have 1:1 orthology with tetrapod genes. With this stated benefit, and with how many *Polypterus senegalus* genes are talked about, we wonder if the authors are missing an opportunity to formalize the gene naming conventions in non-teleost actinopterygians. For example, there are cases where gene names have already been assigned a name in the current RefSeq annotation of *Polypterus senegalus*, such as *pax7a* and *col17a1b*, and the authors keep these names. However, the “a” and “b” copy naming convention comes from the zebrafish, in which the ancestral gene was duplicated in the tsWGD, and thus the *Polypterus senegalus* is not expected to possess a *pax7b* or *col17a1a*. Meanwhile, the authors newly annotate at least 18 genes that were previously only given LOC IDs, and for these genes they assign tetrapod-like gene names, i.e., no “a” or “b” versions. However, in the case of the manually annotated genes *esam*, *tgfb1*, *fn1*, *fgf10*, *lima1*, and *tnc*, zebrafish have “a” and “b” versions of each of them. Therefore, if the goal is to keep consistency with the naming convention of already annotated genes, one of these versions would be selected and used as the gene ID, e.g., *esama* or *tgfb1b*. While it is difficult to challenge already

assigned gene IDs, the discussion of so many different genes in the paper, with many being annotated for the first time, could make this a good opportunity to establish names in the *Polypterus senegalus* that make orthology with the axolotl and other tetrapods clearer, and thus further highlight the benefit of having fish models that are not teleosts.

Additionally, the axolotl gene nomenclature is not following the generally agreed upon style of the salamander community (<https://doi.org/10.1002/dvdy.351>), and so should be changed. The current style in the paper also makes it sometimes unclear if what is being discussed is a protein or a gene due to not using italics for axolotl genes.

We thank the reviewer for raising this point. We have now revised the manuscript by correctly naming *Polypterus* genes that do not have WGD-derived paralogs and assigning orthology/annotation to several LOC ID genes for both *Polypterus* and axolotl in the Supplementary Data 10, which features all gene names and our assigned annotation. Furthermore, we thank the reviewer for suggesting the citation for proper salamander gene nomenclature. We have corrected axolotl gene/protein names throughout the manuscript accordingly.

7) Is there a reason that the authors do not investigate the spatial location of pro-inflammatory and anti-inflammatory genes in their spatial transcriptomic data? They show that in the snRNA-Seq data, these genes are expressed by different populations of cells, and thus it would be interesting, and potentially informative, to see if these different populations are also located in different areas of the regenerating fin.

We appreciate this reviewer's question. Unfortunately, spatial transcriptomics is limited in terms of detection of transcripts of moderate to low abundance or those expressed by few cells. As a result, most transcripts we identified as pro- or anti-inflammatory markers of myeloid cells, while readily detected in single nucleus RNA-seq, were below detection level of 10x spatial transcriptomics. Nevertheless a few genes (*spp1* and *clec7a-like*) did show expression and are now featured in the revised Fig. 10, providing further insight into the timing and location of pro- (*spp1*) and anti-inflammatory (*clec7a-like*) responses.

8) In Figure 9, it is not really clear why the genes discussed here stood out or were selected for further examination. Could be better if the authors stated something like that *tfpi2* was selected for study because they were interested in seeing what genes were specific to *Polypterus* regeneration. Conversely *tnfaip6* seems to have been investigated because it's expressed in the mesenchymal cells of amniotes and teleosts, demonstrating a strong conservation of its role over vertebrate evolution. By more explicitly stating the reasons that drove the selection of these genes, it could enable the authors to more clearly highlight the benefits of studying regeneration across species, and to later discuss the ways in which future experiments are impacted by the knowledge of if gene roles in a particular species are derived or conserved.

We thank the reviewer for encouraging us to dive into a more exploratory analysis of our datasets. This section was removed and expression of wound epidermis and blastema genes is more thoroughly explored in the sections entitled "Reactivation of apical ectodermal ridge-like gene programs in the *Polypterus* wound epidermis" (starting at line 225), "Blastema cell heterogeneity" (starting at line 299) and "Emergence of proximal-distal organization within regenerating connective tissue" (starting at line 336).

Minor comments:

1) In Figure 1c, a color-coded correspondence of structures between species could be very helpful for people not familiar with the species. For example: in the endoskeleton of the *Polypterus senegalus* fin what corresponds to humerus, radius, carpals, phalanges, etc. in the axolotl.

We appreciate this reviewer's suggestion. Unfortunately, direct correspondence (i.e.: bone-to-bone) of endoskeletal elements in ray-finned fish (*Polypterus*) and tetrapods is unresolved and remains an area of active discussion in comparative morphology and evolutionary developmental biology. While the *Polypterus* fin contains proximal elements that are often referred to as "propterygium," "mesopterygium," and "metapterygium," these structures do not have unequivocal homologs to specific limb bones such as the humerus, radius, or ulna (see e.g. Coates, 1994 PMID: 7579518.; Boisvert et al., 2008

<https://doi.org/10.1038/nature07339>; Yano and Tamura 2012 <https://doi.org/10.1111/j.1469-7580.2012.01491.x>). The evolutionary origin of the tetrapod limb endoskeleton likely involved elaboration and regionalization of the metapterygial axis, but even this interpretation is not universally accepted. If this reviewer does not oppose it, we prefer to maintain the figure panel as is.

2) There is a reference to figure (Supplementary Fig. 4c and d), however, there is no “d” panel

We agree with the reviewer. Issue has now been corrected – this figure has been removed.

3) Page 9: “Contrary to mammals, urodele amphibians such as newts can regrow severed limbs even at high environmental oxygen concentrations.”

This sentence makes it sound like mammals can regrow severed limbs.

We agree with the reviewer. Issue has now been corrected. Sentence now reads: “Urodele amphibians such as newts are capable of regenerating severed limbs, even under high environmental oxygen concentrations.” (line 642)

4) Page 8: “Unlike *dif1bb*, *lima1* and its axolotl ortholog LIMA1 were expressed... (Fig. 10d).” *dif1bb* is the wrong gene to say here. The figure states “DIP2B”, and *dif1bb* is not mentioned anywhere else in the paper.

The reviewer is correct. Issue fixed. Sentence now reads: “Unlike *dip2b*, *lima1* and its axolotl ortholog *Lima1*...”. (line 547)

5) Page 6: “A previous study showed that among *Polypterus* myoglobin-like proteins, PseMb1 the product of LOC120537046, termed PseMb1,”

The phrasing of how PseMb1 was given its name should be improved.

We agree and have revised this sentence for clarity. Sentence now reads: “A previous study showed that among *Polypterus* myoglobin-like proteins, PseMb1 — encoded by *LOC120537046* — exhibited the highest sequence identity to the mammalian *MB* gene.” (line 442).

6) It is not clear if the authors imply that previous studies which found that *Frem2* is a marker of wound epidermis cells in the axolotl limb have assigned the gene ID incorrectly, and that it is actually *Frem3*. It would be good if the authors are clearer about their thoughts on axolotl *Frem2*/*Frem3*, and how its expression compares to *Polypterus*.

We thank the reviewer for pointing out this issue. We have now addressed this in greater depth in the section entitled “Reactivation of apical ectodermal ridge-like gene programs in the *Polypterus* wound epidermis”. (starting at line 225). Furthermore, expression dynamics of *frem2/frem3* genes in *Polypterus*, axolotl and zebrafish are now shown in Supplementary Fig. 7.

7) A comment on comparing single-nucleus transcription (de novo transcription) to spatial transcriptomics (whole cell transcripts), would be appreciated.

We agree with the reviewer that addressing the technical differences in transcription detection between snRNA-seq and spatial transcriptomics is important, as these may help explain the differences in the timing of expression peaks we observed for certain genes. To address this, we added the following text (starting at line 236) in the section entitled “Reactivation of apical ectodermal ridge-like gene programs in the *Polypterus* wound epidermis”: “We observed a difference in the timing of the expression peak for basal epithelial markers such as *col17a1* and others (*cyp27b1* and *krt17* – detailed below) in our snRNA-seq data compared to our spatial transcriptomics data (Fig. 4a and b). Specifically, these markers showed their highest expression levels at 1 dpa in the snRNA-seq and at 3 dpa in the spatial transcriptomics dataset. The intrinsic technical differences between snRNA-seq, which captures newly synthesized nuclear transcripts, and spatial transcriptomics, which primarily reflects accumulated cytoplasmic mRNAs, may explain these differences.”

8) The authors use the word integration of data across the manuscript to refer to how they use sn-sequencing data and spatial transcriptomics. This could be misleading as the data was not integrated but rather it was analysed side-by-side.

We agree with the reviewer and have now removed the word integration from such contexts throughout the text and title.

Reviewer #3 (Remarks to the Author):

This was an interesting paper that establishes the fish *Polypterus senegalus* as a new model for studying regeneration in the fin. Although zebrafish have been used for this purpose, only the dermal rays of the zebrafish fin can regenerate, which are different from the endochondral bones present in limbs. *Polypterus* fins, on the other hand, have more extensive endochondral bones that are capable of regenerating, making them a better model for comparative studies of fin and limb regeneration. Here, the authors use a variety of multi-omics approaches, including single-nucleus RNA sequencing, spatial transcriptomics, and ATAC-sequencing to identify commonalities and differences between regeneration in axolotl limbs and *Polypterus* fins.

Overall, the contribution of a new organism for studying the process of regeneration is significant. The authors present several impressive datasets and make great strides in establishing *Polypterus* as a useful model. Although many of the genes and pathways identified have been shown in other systems, the authors do a good job of explaining the benefits of studying regeneration in *Polypterus* and identify some unique genes and pathways that warrant further study in the future. There are no concerns about the methodology, and this is an impressive amount of data. However, the context for the extensive results was sometimes lacking. This made it difficult to understand the conclusions the authors were drawing and to assess the validity of these conclusions. Please see some minor clarifications to address below.

Minor clarifications:

1. Please label the figures (Fig.1, Fig. 2, etc). It was difficult to toggle back and forth between the manuscript, figure legends, and figures when the figures had no labels. It would also be helpful to have the lines numbered in the manuscript.

Figures have now been labelled, and line numbers have been added.

2. An issue that was recurrent throughout the manuscript was a lack of context for the findings. Many sections in the results section could use at least one clarifying sentence. Please see some select examples below.

We thank the reviewer for bringing up this issue. We have now made a conscious effort to add concluding sentences to all sections.

3. In the first results section (“the cellular landscape of *Polypterus* fin regeneration”) the authors mention that erythrocytes are not important in *Polypterus* for fin regeneration while they are in salamanders. Are erythrocytes also not important in zebrafish fin regeneration and therefore the role of erythrocytes is limb specific? In the discussion section, the authors state that this warrants further study, but more discussion about this would be useful.

We thank this and other reviewers who raised the issue of the need for a more in-depth comparative analysis of these cells across taxa. We are restating here our response to reviewer#1, which addresses this matter: “We now have analyzed the recruitment of these cells in *Polypterus*, axolotl and zebrafish and added a results section dedicated to this matter, entitled: “Limb and fin regeneration followed by endoskeleton-level amputation is marked by robust expansion of *hif4a*⁺ erythrocytes”. (starting at line 369). We show that the emergence of these cells is not a hallmark of zebrafish tail fin ray regeneration. Our analysis of the top 100 differentially expressed genes in erythrocytes of *Polypterus* and axolotl revealed confirmed the overall lack of ECM remodeler or secreted morphogen expression and showed

typical as well as atypical erythrocyte genes commonly and uniquely expressed in these species. A careful reassessment of these datasets revealed the erythrocyte-specific enrichment of a HIFA paralog and phylogenetic analysis supports that these genes encode HIF4A ortholog. The presence of *Hif4a*-expressing erythrocytes during regeneration in *Polypterus* and axolotl—but not zebrafish tail fin ray—possibly consists in a unique adaptation to the metabolic and vascular demands of regenerating an endoskeleton-bearing appendage. These analyses are shown in a new figure (Fig. 8), new supplementary data (Supplementary Fig. 8) and table (Supplementary Data 8).”

4. For the next section (“cellular contributions to the *Polypterus* fin wound epidermis and blastema”) there is a very nice and thorough description of the role of connective tissue cells in the *Polypterus* fin blastema, but no context provided. Is this similar in other fish and limbs? Or different?

We thank the reviewer for the constructive comment and for suggesting that we provide additional context to the data presented in this section. To address this and related concerns raised by other reviewers regarding our description of *col6a3* as a marker of connective tissue contribution to the blastema, we have restructured the relevant section. In the revised manuscript, we now include two new sections focusing on the blastema compartment (“Blastema cell heterogeneity – line 299” and “Emergence of proximal-distal organization within regenerating connective tissue – line 336”).

To provide context, in the section “Blastema cell heterogeneity,” we have added references supporting the statement that connective tissue cells are major contributors to the blastema in both axolotl and zebrafish. In the following section, “Emergence of proximal-distal organization within regenerating connective tissue” we combine spatial transcriptomics and sn/scRNA-seq analyses to demonstrate a proximal–distal subdivision of connective tissue associated with the blastema region in both *Polypterus* and axolotl at later stages of regeneration. Cell populations corresponding to these distinct subdomains—identified by expression of specific marker genes—were also detected in a zebrafish fin regeneration snRNA-seq dataset. Together, these results indicate that specialization of connective tissue within the blastema occurs in a similar manner in regenerating fish fins and axolotl limbs.

5. I thought that the sections about hypoxia and the DNA damage response were the clearest to understand. The authors did a good job of describing the role of both pathways in other organisms and clearly identify *Polypterus* specific pathways and genes that warrant more study in the future.

We thank the reviewer for appreciating how this section was presented.

6. In the section “epigenetic analysis uncovers candidate fin regeneration-responsive elements” it was unclear if the authors are hypothesizing that Peak 5 is driving *lima1* expression instead of *dip2bb*. Please state this clearly if that is the case. For example, for the sentence “While both genes are upregulated over 2-fold, bulk RNA-seq showed that *lima1* expression at 3 dpa is over 15 times greater than that of *dip2bb*” please add a sentence such as “indicating that Peak 5 is likely driving expression of *lima1*.” If that conclusion is not correct, please state what the correct conclusion is from this section about *dip2bb* and *lima1*.

We thank the reviewer for raising this issue and we have now modified this statement, which reads as follows: “While further experiments will be necessary to confirm the gene under peak 5 influence, expression of *lima1* seems more associated with progression of regeneration. Both genes were upregulated by just over 2-fold, bulk RNA-seq showed that *lima1* expression at 3 dpa is over 15 times greater than that of *dip2bb*” (line 539).

7. It would be very helpful to have some sort of table identifying which genes and/or pathways are conserved and diverged between *Polypterus*, salamanders, and other fish. Do the authors think that regeneration is more conserved or diverged between *Polypterus* and salamanders? What about between *Polypterus* and zebrafish? Many genes and pathways were discussed, and it was difficult to get a sense of how conserved this process is between the different organisms.

We understand and agree with this reviewer's request for clarification on this matter. We have now included in Fig. 11 a table that provides a schematic overview of shared and species-specific mechanisms and characteristics of limb and fin regeneration in axolotl, *Polypterus*, and zebrafish, as concluded from our comparative analysis.

Reviewer #4 (Remarks to the Author):

In the work from Sousa et al., the authors leverage evolutionary relationships across the vertebrate tree of life to compare limb regeneration by the Urodele axolotl salamander with that of a more closely related polypterid, *Polypterus senegalus*. The endoskeletal fin regeneration of *Polypterus* is a closer comparison than teleost regeneration of dermal bony fins. This manuscript builds on the Schneider lab's prior work establishing *Polypterus*' regenerative ability and evolutionary relationship relative to axolotl and other fish lineages (Darnet, et al., 2019, PNAS). In this manuscript, the authors perform a full multi-omic comparison of axolotl limb and *Polypterus* fin regeneration that encompasses single nuclei RNA-seq (snRNA-seq), Visium spatial transcriptomics at 3 days post amputation (3dpa), and bulk Assay for Transposase-Accessible Chromatin sequencing (ATAC-seq). The authors identify a number of similarities and differences in the transcriptional and genomic regulation immediately following amputation. The rigor associated with having both limb and fin tissues subjected to the same experimental methodologies and analyses is a great resource for the regeneration field and represents an excellent standard for future studies.

Despite the technical standard that this work represents, I struggled to find cohesive conclusions from what seemed like observational "vignettes" picked from the multi-omic datasets. I also was unclear if, or how, this work related to their prior analysis of limb and fin regeneration (Darnet et al.). I don't think this should detract from the work, but I do think there are a few meaningful changes that could create a more focused and presentable manuscript.

- My primary critique is that this work should be presented primarily as a technical resource since none of the individual results sub-sections are comprehensive or mechanistic enough to make major conclusions. In that vein, I would suggest that the authors put effort into making the data accessible to readers via an explorable R-Shiny site or uploading data to the Broad Institute Single Cell Browser (or equivalent). This would translate the technical prowess in the manuscript to a highly impactful resource for the regeneration community.

We appreciate the reviewer's suggestion and have now deposited and made immediately available our *Polypterus* single nucleus RNA-seq experiment in the Broad Institute Single Cell Browser.(see Data Availability, line 908).

- One of the primary arguments put forth by the authors for using the *Polypterus* fin as a comparative model to the axolotl limb, in contrast to the more ubiquitous zebrafish fin, is the presence of endoskeletal regeneration. Yet, the authors make no connections between regeneration of the fin skeleton and axolotl limb bones. This seems like a large oversight, and it is not clear if it is because of technical limitations such as not being able to detect skeletal cell types in snRNA-seq data (no clusters were described). I recognize that the timepoints the authors examined are early after amputation and do not represent timepoints that skeletal patterning is happening, however a comparison of skeletal cell types and signals that influence processes like bone histolysis, and skeletal progenitor signaling could be examined even at early phases.

We understand and agree that this issue needs clarification. We had originally referred to our *Polypterus* experiments as endoskeleton regeneration, while our intent was to place emphasis on the region where the amputation is being made – not necessarily the endoskeletal bones. The proximal fin region bears the bones that are homologous to tetrapod limb bones, but it also contains the cellular complexity seen in limbs. This cellular complexity (i.e.: presence of muscle, large and diverse connective tissue cell populations, tendons, cartilaginous joints), is lacking in the fin ray region and hence, cannot be assessed by studying zebrafish fin ray regeneration. To clarify this, we changed wording in the introduction and

throughout the text to emphasize this point and to refer to our approach as a proximal fin amputation at the level of the endoskeleton. (paragraph starting at line 100).

Nevertheless, the reviewer's comment prompted us to examine in more detail our datasets to evaluate genes associated with endoskeletal or fin ray progenitors. With the addition of spatial transcriptomics datasets for later stages of *Polypterus* fin and axolotl limb regeneration, we were able to single out endochondral and fin ray regeneration regions in the *Polypterus* fin at 7 dpa. We have now added a new figure (Fig. 3) and a differential gene expression analysis of the endochondral and fin ray regions (supplementary data 1) that, we believe, addresses the reviewer's concerns. These analyses are now reported in a new section entitled: "Distinct regeneration programs of endochondral and dermal skeleton in *Polypterus*." (starting at line 185).

- Related to the point above, the Discussion section of the manuscript is used primarily as a re-hash of result subsections rather than an opportunity in the manuscript to describe limitations, future outlook/directions, and postulating on a more cohesive overview of the work (e.g., in what ways is *Polypterus* regeneration more similar to teleost regeneration and what ways is it more like limb regeneration?). I appreciate that the authors try to put some of their results in context of prior literature, but it is sometimes hard to follow with non-precise reference to other papers. For example, the connection of *Kazald* expression to other systems was not very clearly articulated and I'm not sure if the reference given really provides the evidence that the authors indicate in the discussion text.

We thank the reviewer for bringing up this issue. We have reworked the discussion section to reduce restatements of findings and to emphasize limitations, future directions, and to frame our results in the context of existing literature.

- As a reader, one of the most interesting and relatively untouched aspects of this work is the difference in spatial distribution of gene expression in the proximal-distal axis between the two organisms. The authors point this out relative to DNA damage markers between *Polypterus* and axolotl, but this seems evident in many other gene comparisons between fin and limb in the manuscript's figures. Taking an unbiased gene set (e.g., top 100 DEGs) and determining the relative proximal-distal expression zone (using some standard of normalization and distance) would be a very interesting and quantitative comparison using spatial transcriptomic data. This would really leverage the author's rigorous data in a way that is novel and unlikely to be represented in other manuscripts.

We thank this and other reviewers for raising this point and prompting us to examine this proximal-distal axis in more detail. We followed this and other reviewer's suggestions and now provide evidence for this PD subdivision with corresponding identifying gene markers. We are restating here our response to reviewer#1, which addresses this matter: "we refined our cluster analysis on spatial transcriptomics data and now provide novel findings regarding the two blastema clusters, which we refer to as proximal and distal CT cell populations. We had initially worked with default parameters for clustering, which led to overclustering of spatial data. We corrected this now by applying within-cluster sum of squares (WCSS) as a metric of clustering compactness. This resulted in fewer clusters that are more likely to be biologically meaningful. Even so, both *Polypterus* 7 dpa and axolotl 14 dpa showed proximal and distal CT compartmentalization with k-means clusters = 5. Analysis of the top 100 genes expressed in these territories revealed clear distinguishing markers and other features of these subdivided cell populations conserved between *Polypterus* and axolotl. These new analyses are described in a new section entitled "Emergence of proximal-distal organization within regenerating connective tissue" (starting at line 336), and featured in newly produced Fig. 7, and Supplementary Data 3.

- Finally, I have a hard time really agreeing on the author's definitions of "wound epidermis" markers derived from snRNA-seq in Figures 1-3. The authors describe "wound epidermis" clusters, but these clusters seem to exist in the uninjured data set as well, just with fewer cells represented relative to post amputation. They describe *col17a1b* as a marker of these epidermal cells, but again, there are *col17a1b*-expressing cells before regeneration as well. It is not until Figure 5 when markers are connected between Visium and the snRNA-seq clusters do you observe marker genes that are expressed specifically after amputation in wound epidermis snRNA-seq clusters. Markers like *col17a1b* should be clarified as upregulated or "amplified" during regeneration instead of markers that uniquely identify the wound

epidermis. In general, I think this double standard for wound epidermis markers is confusing and I would simply suggest moving the *col17a1b* data to supplemental data since the data in Figure 5 is much more convincing although it lacks high resolution HCR micrographs.

We thank this and other reviewers for drawing our attention to the matter of *col17a1* expression in our study. This has prompted us to generate the additional spatial transcriptomics datasets for both species, as well as incorporating analyses of publicly available axolotl limb regeneration and zebrafish tail fin regeneration single cell/nucleus RNA-seq datasets, which greatly enriched our study. We restate here the response provided to reviewer #1: “A more in-depth analysis of wound epidermis cells and candidate markers is now reported in results section entitled “Reactivation of apical ectodermal ridge–like gene programs in the *Polypterus* wound epidermis”. (starting at line 225). Whereas *col17a1* does become upregulated in the distal wound epidermis, it is expressed in homeostasis and throughout regeneration in all species and therefore is now more accurately described in our manuscript as a basal epidermis marker. A refined analysis of genes differentially upregulated in the presumptive wound epidermis is now provided and shows a more established marker of activated keratinocytes (*krt17*) being conspicuously upregulated in the distal wound epidermis in both single nucleus RNA-seq and spatial transcriptomics datasets. HCR-FISH for *krt17* was now added and largely mirrors the pattern we observe in our *Polypterus* spatial transcriptomics data. Additional markers including *cyp27b1* and *mmp13* are also shown, as are comparisons of gene expression patterns with axolotl and zebrafish. These data resulted in new figure and tables, specifically: Fig. 4 and 5, Supplementary Figs. 5, 6 and 7, and Supplementary Data 2.